# Tumor-induced orexigenic imbalance lowers protein appetite and drives early organ wasting symptoms

Afroditi Petsakou [1,2] ✉, Elizabeth Filine [1], Matthew Li [1], Yuchen Chen[1], Alice Zheng [1] & Norbert Perrimon [1,3] ✉

Cancer cachexia (CC) is characterized by organ wasting and ensuing involuntary weight loss. Despite advances, underlying mechanisms initiating CC remain unclear, including early symptoms like anorexia. Here, we use a fly gut-tumor model with a precise time-window before organ wasting starts. We show that tumor-induced factors involved in inflammation (unpaired 3/ Interleukin-6-like) and reduced insulin signaling (ImpL2/ Insulin Growth Factor Binding Protein) decrease NPF (Neuropeptide F/ Neuropeptide Y) in the brain prior to organ wasting. This early NPF decrease triggers low protein-specific food appetite and anorexia. We find that ImpL2 reduces NPF signaling while upd3 helps by concurrently affecting the blood brain barrier. Tumor-induced NPF decrease, and early reduction of protein appetite drive the onset of weight loss and exacerbate the risk of death during organ wasting. Altogether, we provide evidence for an early orexigenic brain imbalance causing low protein appetite that regulates the onset and outcome of organ wasting.

Cancer cachexia (CC) is a devastating wasting syndrome, developed in late cancer stages. During CC, reduced food intake - together with elevated energy expenditure, systemic inflammation, insulin resistance, and heightened catabolic activity- leads to the loss of muscle and fat at a rate that exceeds their replacement[1]. This multi-organ dysfunction causes involuntary weight loss and wasting of muscle and adipose tissue[1]. Recent advances have identified metabolic, inflammatory, and neuronal networks associated with CC, highlighting how conserved intertwined pathways drive diverse aspects of organ wasting (e.g., increased lipolysis, reduced insulin signaling, muscle catabolism)[1]. Despite these advances, mechanisms responsible for the onset of CC (organ wasting and weight loss) remain unclear. Organ wasting is irreversible, has no cure, and greatly diminishes the quality of life for cancer patients. About 50–80% of advanced cancer patients develop CC. As a result, there is a pressing need for early diagnosis and mitigating multimodal therapeutic interventions to slow down the onset of CC and improve the quality of life for CC patients[1,2].

Early signs of CC include anorexia (loss of appetite), which precedes weight loss and organ wasting[1,3]. In addition, CC patients regularly experience changes in food preferences[4,5]. While malnourished CC patients have low survival rates compared to well-nourished patients[6], the etiology of anorexia and appetite changes during CC remain understudied, including whether feeding-related pathways impact the advent of CC[7,8]. These outstanding questions have proven difficult to address in murine models due to the complexity and onset variability of CC. We therefore reasoned that a simpler model like *Drosophila*, which has a defined time window for organ wasting[9–11], a comprehensive understanding of feeding-related signaling pathways[12] and an advanced genetic toolset[13], would be ideal to identify conserved primary mechanisms that help early diagnose and mitigate CC.

The *Drosophila* gut has emerged as an attractive adult-specific gut organ wasting model due to the precise control for the onset of organ wasting[9–11]. In this model, organ wasting is triggered by the activation of the Yorkie (Yki)/Yap oncogene, which is expressed in intestinal progenitor cells using the *escargot*-Gal4 driver, together with the

[1]Department of Genetics, Harvard Medical School, Boston, MA, USA. [2]Department of Developmental and Molecular Biology, Albert Einstein College of Medicine, Bronx, NY, USA. [3]Howard Hughes Medical Institute, Boston, MA, USA. ✉e-mail: Afroditi.Petsakou@einsteinmed.edu; perrimon@genetics.med.harvard.edu

temperature-sensitive Gal4 repressor *Tubulin*-Gal80[TS] (referred as *esg[TS] > yki[act]*)[9]. Specifically, *esg[TS] > yki[act]* flies are reared during development at low temperatures (18 °C) to inhibit oncogene activation in the gut. When *esg[TS] > yki[act]* flies reach adulthood, the temperature is then raised (29 °C) to allow for Yki activation in the gut (defined as day 0, d0)[9]. A week of gut Yki-tumor induction in adult flies leads to the onset of organ wasting symptoms, such as increased lipolysis, reduced insulin signaling, bloating (ascites-like symptom) and reproductive organ atrophy, followed by fat cell degeneration and myofiber degradation[9–11,14], reminiscent of gut CC symptoms.

Two of the most highly expressed tumor-induced secreted factors in *esg[TS] > yki[act]* flies are the conserved inflammatory cytokine upd3 (unpaired 3) and insulin antagonist ImpL2 (imaginal morphogenesis protein-late 2)[9,14], whose mammalian counterparts, IL-6 (interleukin-6-like) and IGFBP (insulin growth factor binding protein), respectively, are associated with CC[1,15–18]. Tumor-induced ImpL2 drives organ wasting symptoms, including reduced insulin signaling without affecting tumor growth[9,19], whereas upd3 promotes gut tumor growth[14], disrupts the blood-brain barrier (BBB)[20], triggers the secretion of additional ImpL2 from the muscle[14], and impairs adipose metabolic function during organ wasting[11].

Moreover, *Drosophila* is a powerful genetic model to identify signaling pathways driving feeding behavior[12]. Several conserved interlinked hunger and satiety signals have been identified[12], as well as mating- and sex-specific mechanisms that differentially regulate feeding in male versus female flies[21–24]. An important hunger-related signal is NPF, a homolog of mammalian orexigenic hormone NPY (neuropeptide Y). NPF is expressed in ~30 neurons in the fly brain from where NPF signaling promotes feeding in both males and females[25,26]. Another vital role of NPF neurons is that they integrate information about the nutritional state of the animal[27–30]. In addition, NPF is found in enteroendocrine cells in the fly gut[31]. NPF from the gut is reported to have distinct physiological functions from the brain, e.g., NPF signaling from the gut promotes sugar satiety and inhibits food intake[23,32].

In this work, we used gut Yki-organ wasting flies and discovered an underlying mechanism in the brain that drives early orexigenic imbalance, weight loss and significantly diminishes the probability of survival. Specifically, our data support that excessive increase in tumor-induced ImpL2 synergistically with upd3 leads to an early decline of neuronal NPF signaling, which in turn causes inadequate protein-specific nutritional choices and anorexia prior to organ wasting. This early protein-specific orexigenic disparity prevents the tumor-compromised animal from meeting its nutritional needs, hence initiating weight loss and ultimately worsening the outcome of organ wasting by increasing the risk of death.

## Results

### Anorexia precedes organ wasting in gut Yki-tumor flies

Previous work using a capillary-based assay to test the feeding behavior of female *esg[TS] > yki[act]* flies reported a subtle decrease in food intake by d6 of tumor growth[9]. Since anorexia is an early symptom of CC, we reexamined the feeding behavior of both male and female *esg[TS] > yki[act]* flies using a more sensitive approach. Specifically, we used an automated behavior apparatus (FlyPAD, fly proboscis and activity detector) that monitors how long individual flies contact the food in real time[33]. The duration of food contact (total duration of activity bouts) significantly correlates with food ingestion and is used to quantify feeding behavior[33]. Using the FlyPAD, we observed a significant reduction in the feeding behavior of *esg[TS] > yki[act]* flies, with both males and females showing a similar decrease in feeding behavior (Fig. 1a and Supplementary Fig. 1a).

The *escargot*-Gal4 driver is expressed subtly in the brain (Fig. 1b and Supplementary Fig. 1b) in addition to the gut, which could complicate the interpretation of the results. Thus, we prevented expression in the brain by utilizing the neuronal Gal4 inhibitor *elav*-Gal80 (Fig. 1b

and Supplementary Fig. 1b). Both male and female flies with *yki[act]* oncogene expression specific to the gut (Fig. 1c and Supplementary Fig. 1c) and not the brain (*elavGal80;esg[TS] > yki[act]*—hereafter referred as Yki[act]), maintained significantly reduced feeding behavior (Fig. 1d and Supplementary Fig. 1d). To further verify that food intake in Yki[act] flies is significantly decreased, we used a dye-based feeding assay, in which a nontoxic non-absorbed blue dye is added to the food, and food consumption is measured by eye-scoring (Fig. 1e and Supplementary Fig. 1e) or by color spectrophotometry of homogenized flies (Fig. 1f and Supplementary Fig. 1f). Indeed, both male and female Yki[act] flies showed low food intake on d6 of gut tumor growth (Fig. 1e, f, and Supplementary Fig. 1e, f), resembling an anorexic phenotype.

Moreover, we used the blue-dye assay to test food intake in male and female flies with gut tumors that do not cause organ wasting, such as activation of the *Raf* oncogene (*elavGal80; esg[TS] > Raf[GOF]*) or knockdown of the *APC* tumor suppressor (*elavGal80; esg[TS] > APC[RNAi]*). Food intake was not significantly affected compared to control in these gut tumor flies (Fig. 1g, h, and Supplementary Fig. 1g, h), indicating that the anorexic behavior exhibited by Yki[act] flies correlates with organ wasting. Finally, we tested whether reduced food intake in male and female Yki[act] flies is impacted by an alternate diet or the microbiome by changing the food recipe or adding antibiotics, respectively. Neither condition prevented male and female Yki[act] flies from consuming significantly less food (Fig. 1i, j, and Supplementary Fig. 1i, j). Together, these data support that Yki[act] flies undergo anorexic behavior, which is linked to gut tumor-induced organ wasting.

Organ wasting symptoms have been reported to start after d6 in Yki-tumor flies[9–11]. For instance, increased lipolysis, which is marked by a significant drop in triglycerides, occurs in gut Yki-tumor flies by d6 in both female[9] and males (Fig. 1k). In addition, as an indicator of tissue atrophy, we measured dry weight of male and female Yki[act] flies and observed a significant reduction after d6 (Figs. 1l, 5a, and Supplementary Fig. 1k). Strikingly, in both male and female Yki[act] flies, the onset of reduced food intake starts on d5 (Fig. 1m, n, and Supplementary Fig. 1l, m). This is one day before organ wasting symptoms appear, signifying that in Yki[act] flies, the anorexic behavior precedes organ wasting, reminiscent of CC-related anorexia.

### Low neuronal NPF signaling drives anorexia in Yki[act] male flies

Next, we sought to identify signaling pathways driving gut tumor-induced anorexia by testing the expression of known feeding-related signals[12] on d5 (onset of anorexia). We observed distinct changes in male versus female Yki[act] flies, for example, the satiety peptide *upd2/leptin*[34] was significantly increased only in females, and not males Yki[act] flies when compared to control (Fig. 2a, Supplementary Fig. 2a, and Supplementary Table 1). This suggests that feeding behavior is regulated by sex-specific mechanisms[21–24] even under tumor conditions. Given that feeding-related networks in females are complex due to post-mating sex peptide signaling and egg production[21,23,35,36], we focused on the anorexic behavior of male Yki[act] flies.

During the onset of anorexia, we observed several feeding-related signals to have variable expression in Yki[act] males, with *CCHamide-2* (*ccha2*) and *NPF* showing the strongest statistical decrease (Fig. 2a and Supplementary Table 1). *Ccha2* has no known mammalian counterpart and has been proposed to act as a nutrient-stimulated satiety indicator[12], albeit there are conflicting reports regarding its function[37–39]. Since low expression of *ccha2* has been linked to starvation conditions[37], it is likely that this decrease marks the internal nutritional scarcity experienced by Yki[act] flies due to reduced food intake at d5.

Moreover, NPF is a well-studied conserved neuropeptide, and low levels of *NPF* expression cause a decrease in associated behaviors, including feeding[23,26,40–42]. Interestingly, NPY, which is the mammalian homolog of NPF, is reduced in cancer anorexia patients and murine models[43–47]. However, its role in CC remains to be determined.

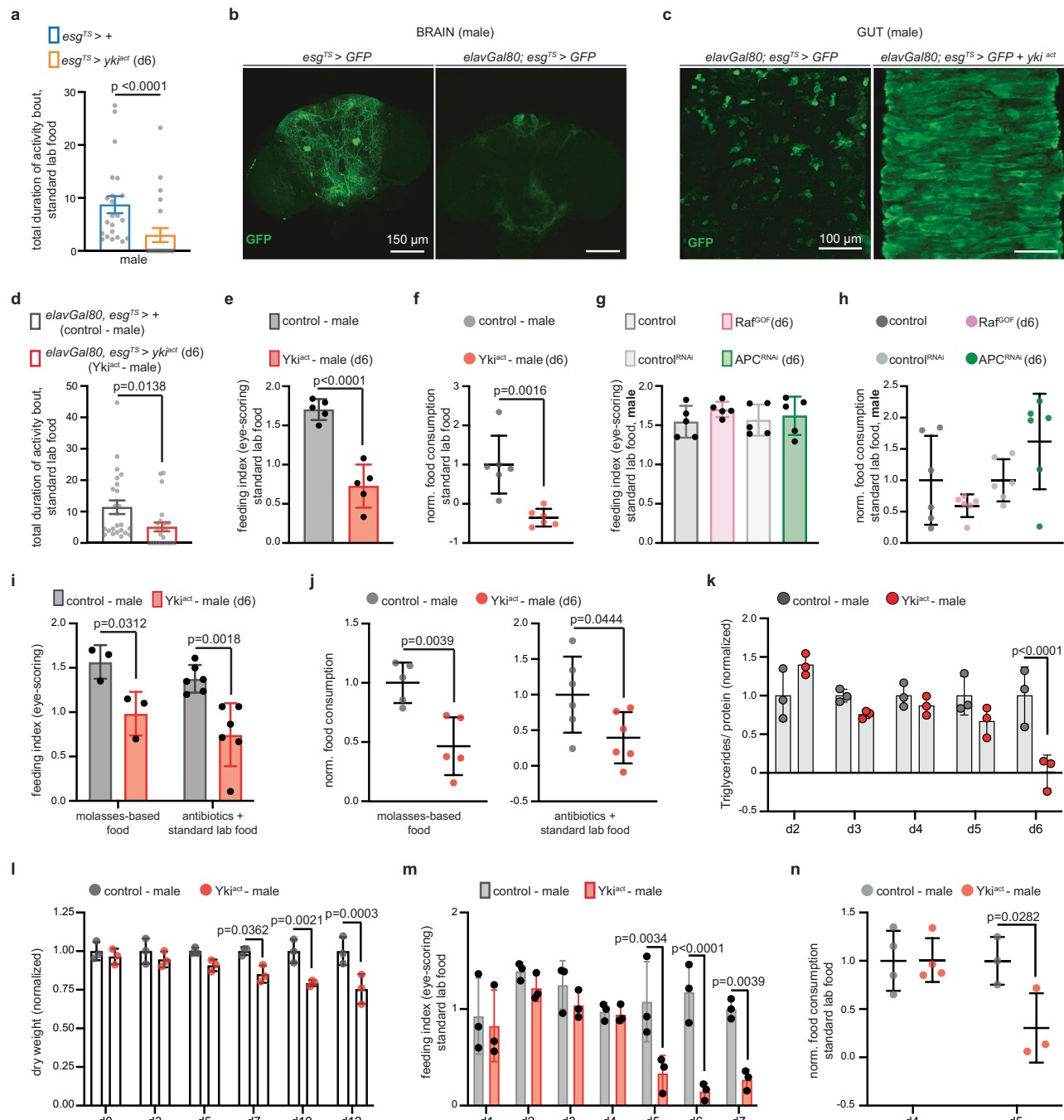

**Fig. 1 | Gut Yki-tumors cause anorexia prior to organ wasting in males. a** FlyPAD assay. *Esg^TS* > + (control, blue) and flies with gut Yki-tumor for 6 days (*esg^TS* > *yki^act*, orange). *n* = 22 male flies per genotype, 4 independent experiments. Two-tailed Mann−Whitney test. **b** Brains from *esg^TS* > *GFP* and *elavGal80;esg^TS* > *GFP* (d6). Two independent experiments. **c** Guts from *elavGal80;esg^TS* > *GFP* and *elavGal80;esg^TS* > *GFP* +*yki^act* (d6). Two independent experiments. **d** FlyPAD assay. Control (gray) and flies with gut-specific Yki-tumor (*elavGal80;esg^TS* > *yki^act*, hereafter: Yki^act, red). *n* = 24 males per genotype (4 independent experiments). Two-tailed Mann−Whitney test. **e** Eye-scoring assay. Flies like (**d**). *n* = 5 biological replicates per genotype (3 independent experiments). Unpaired two-tailed t-test (*t* = 7.152, df = 8). **f** Blue-dye (spectrophotometry) assay. Flies like (**d**). *n* = 6 biological replicates per genotype (3 independent experiments). Unpaired two-tailed t-test (*t* = 4.298, df = 10). **g** Eye-scoring assay. Control (gray), Raf^GOF (*elavGal80;esg^TS* > *Raf^GOF*, pink), control^RNAi (*elavGal80;esg^TS* > *Luciferase^RNAi*, light gray), APC^RNAi (*elavGal80;esg^TS* > *APC^RNAi*, green). *n* = 5 biological replicates per genotype (3 independent experiments). One-way Anova (Tukey's test). **h** Blue-dye (spectrophotometry) assay. Flies like (**g**). Raf^GOF normalized to control, APC^RNAi to control^RNAi. *n* = 6 biological replicates per genotype (3 independent experiments).

One-way Anova (Tukey's test). **i** Eye-scoring assay. Flies like (**d**). Biological replicates per genotype: *n* = 3 (molasses), *n* = 6 (antibiotics). Three independent experiments; two-way Anova (Sidak's test). **j** Blue-dye (spectrophotometry) assay. Genotypes like (**d**). Normalized to control. Biological replicates per genotype: *n* = 5 (molasses), *n* = 6 (antibiotics). Unpaired two-tailed t-test (molasses: *t* = 4.009, df = 8; antibiotics: *t* = 2.298, df = 10). Three independent experiments. **k** Triglyceride assay. Genotypes like (**d**), d2−d6. Normalized to control. *n* = 3 biological replicates per genotype (3 independent experiments). Two-way Anova (Sidak's test). **l** Dry weight assay. Genotypes like (**d**), d0−d12. *n* = 3 biological replicates per genotype, time-point (3 independent experiments). normalized to control. Two-way Anova (Sidak's test). **m** Eye-scoring assay. Genotypes like (**d**), d1−d7. *n* = 3 biological replicates per genotype, timepoint (3 independent experiments). Two-way Anova (Sidak's test). **n** Blue-dye (spectrophotometry) assay. Genotypes like (**d**), d4−d5. normalized to control. Biological replicates per genotype: *n* = 4 (d4); *n* = 3 (d5). Two-way Anova (Sidak's test); 3 independent experiments. Activity bout: seconds. anti-GFP: green. Scale: 150 μm (**b**), 100 μm (**c**). Mean ± SEM (**a**, **d**), SD (**e**−**n**). Exact *p*-values are shown. Source data are provided as Source data file.

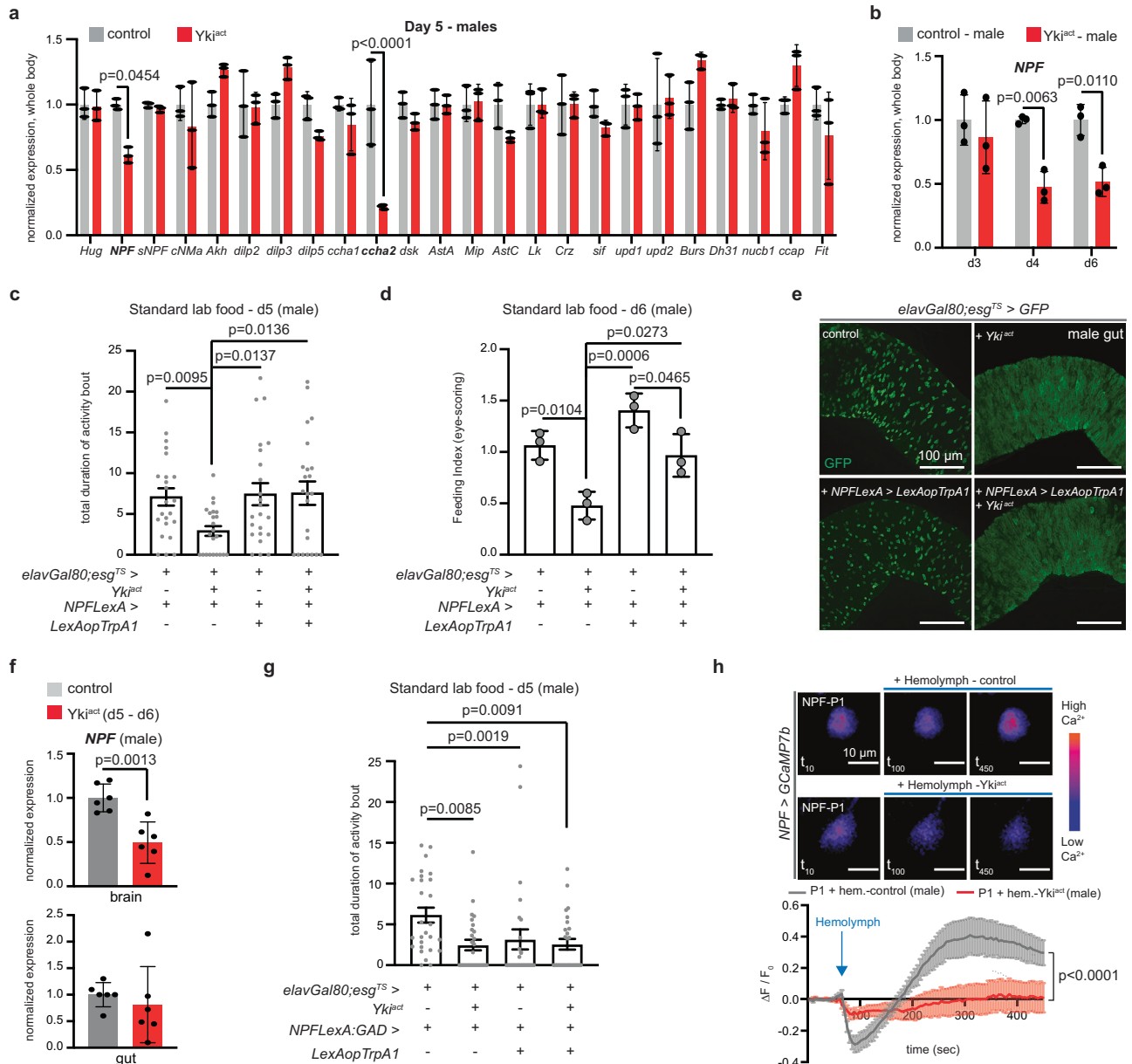

**Fig. 2 | Early brain NPF decrease drives anorexia in Ykiact males. a** Expression levels from the whole body of male control (gray) and Ykiact (red) on d5. *n* = 3 biological replicates per genotype. Normalized to control. Two-way Anova (Sidak's test). *Hug Huggin, NPF Neuropeptide F, sNPF short Neuropeptide F, CNMa CNMamide, Akh Adipokinetic hormone, dilp2 Insulin-like peptide 2, dilp3 Insulin-like peptide 3, dilp5 Insulin-like peptide 5, ccha1 CCHamide, ccha2 CCHamide2, dsk drosulfakinin, AstA Allatostatin A, mip myoinhibitory, AstC Allatostatin C, Lk Leucokinin, Crz Corazonin, sif SIFamide, upd1 unpaired1, upd2 unpaired2, Burs Bursicon, Dh31 diuretic hormone 31, nucb1 nucleobindin-1, ccap Crustacean cardioactive peptide, Fit female-specific independent of transformer.* **b** *NPF* levels. Genotypes like (**a**) on d3, d4, d6. Normalized to control. *n* = 3 biological replicates per genotype, timepoint. Three independent experiments. Two-way Anova Sidak's test (Supplementary Table 1). **c** FlyPAD assay. Control (*elavGal80;esgTS/+; NPFLexA/+*), Ykiact (*elavGal80;esgTS > ykiact + NPFLexA > +*), flies with increased NPF signaling (*NPFLexA > LexAopTrpA1 + elavGal80;esgTS > +*), flies with increased NPF signaling and Yki gut-tumor (*NPFLexA > LexAopTrpA1 + elavGal80;esgTS > ykiact*) on d5. *n* = 23 flies per genotype. Four independent experiments. Kruskal–Wallis Dunn's test.

**d** Eye-scoring assay. Genotypes like (**c**) on d6. *n* = 3 biological replicates per genotype. Three independent experiments. One-way Anova (Tukey's test). **e** Male guts expressing GFP in progenitor cells. Control (*elavGal80;esgTS > GFP; NPFLexA > +*). Two independent experiments. Anti-GFP: green. scale bar: 100 μm. (GFP quantifications: Supplementary Fig. 2b.) **f** *NPF* levels in brain and gut. Genotypes like (**a**) d5–d6. *n* = 6 biological replicates per tissue, genotype. Four independent experiments. Normalized to control. Unpaired two-tailed t-test (brain: *t* = 4.404, df = 10, gut: *t* = 0.6116, df = 10). **g** FlyPAD assay (d5). Control (*elavGal80; esgTS/+; NPFLexA:GAD/+*). *n* = 27 flies per genotype. 5 independent experiments, Kruskal–Wallis Dunn's test. **h** Color-coded frames of NPF-P1 neurons from *NPF > GCAMP7b* male brains before ($t_{10}$) and after ($t_{100}$), ($t_{450}$) hemolymph. Supplementary Movies 1 and 2. Graph shows average fluorescence intensity ($\Delta F/F_0$) per frame (5 s/frame). Control hemolymph: gray. Ykiact d6 hemolymph: red. *n* = 5 neurons per condition. 3 independent experiments. Two-way Anova (Tukey's test). Individual $\Delta F/F_0$: Supplementary Fig. 2e. Scale bar: 10 μm. Mean ± SEM (**c**, **g**, **h**), SD (**a**, **b**, **d**, **f**). Activity bout: seconds. Exact *p*-values are shown. Source data are provided as Source data file.

Additionally, reduction of *NPF* expression was observed in an eye-tumor organ wasting model in the fly, but the role of NPF was not explored[44]. In that study, changes in food intake were associated with a decrease in the orexigenic *sNPF* (*short Neuropeptide F*) and an increase in the anorexigenic *nucleobinding1* (*nucb1*)[44]. *sNPF* and *nucb1* expression showed no difference in gut-tumor Yki[act] flies when compared to control (Fig. 2a and Supplementary Fig. 2a), suggesting that feeding behavior is regulated by distinct mechanisms in different tumor models. Since the function of NPF in tumor-induced anorexia and organ wasting remains unclear, we explored its role in the anorexic behavior of male Yki[act] flies.

We observed that by d4 of gut tumor induction, Yki[act] male flies undergo more than 40% decrease in *NPF* expression compared to control (Fig. 2b), indicating that a large reduction in NPF signaling could be a very early inducer of anorexia. To test this, we systemically increased NPF signaling by expressing the thermosensitive cation channel TrpA1 with the *NPF-LexA* driver (*NPFLexA > LexAopTrpA1*). Specifically, at high temperature (~29 °C), TrpA1 allows the influx of cations inside the cell, which increases activation and release of NPF from NPF-expressing neurons and enteroendocrine gut cells[23,26,32,40,41]. A concurrent increase in NPF signaling and Yki-tumor growth (*elav-Gal80; esg[TS] > yki[act] + NPFLexA > LexAopTrpA1*) in male flies was sufficient to rescue anorexia on d5 and d6 (Fig. 2c, d). The TrpA1-induced increase of NPF signaling did not change the growth of gut Yki-tumor (Fig. 2e and Supplementary Fig. 2b). This suggests that the rescue of anorexia is directly linked to increased NPF signaling and is not an indirect effect due to changes in the development of tumor growth.

Moreover, *NPF* expression is significantly reduced in the brain of male Yki[act] flies compared to control (>40% decrease), but is variable in the gut (Fig. 2f). We therefore reasoned that anorexia likely originates from the reduction of *NPF* levels in the brain and not the gut. To test this, we used the *NPF-LexA:GAD* driver[48] that is repressed by the neuronal inhibitor *elav*-Gal80, thus only allowing NPF signaling from the gut. TrpA1-induced elevated NPF signaling solely from the gut and not the brain of Yki-tumor flies (*elavGal80;esg[TS] > yki[act] + NPFLexA:GAD > LexAopTrpA1*) did not rescue the anorexic behavior at d6 (Fig. 2g), supporting that low *NPF* levels from the brain drive anorexia. In addition, flies with TrpA1-induced activation of NPF solely from the gut showed reduced food intake (Fig. 2g), in agreement with previous reports that NPF signaling from the gut has distinct roles from the brain[23,32].

Neuronal silencing leads to reduction in neuropeptide expression[49,50] and *NPF* expression levels are prone to change due to external factors[51]; we therefore speculated that low *NPF* expression in Yki[act] male flies is the result of tumor-derived signals that silence NPF neurons in the brain. To test this, we performed an ex vivo brain assay where we assayed the response of NPF neurons in the presence of control and tumor-derived secreted factors (Supplementary Fig. 2c). Specifically, we expressed the genetically encoded Ca[2+] indicator GCAMP7b with the *NPF-Gal4* driver (*NPF > GCAMP7b*) in the brain and imaged NPF neurons while adding hemolymph (fly blood) from control or Yki[act] male flies (Supplementary Fig. 2c). We focused on the dorsal median NPF-P1 neurons (Supplementary Fig. 2d), as they are proposed to be indicators of internal state imbalance[30] and are selectively activated by both hunger and food-odor attractiveness[27]. Interestingly, adding control hemolymph to NPF-P1 neurons caused a post-inhibitory excitation response. This excitation response indicates that secreted factors in control hemolymph allow neuronal activation (Fig. 2h, Supplementary Fig. 2e, and Supplementary Movie 1). In contrast, when we added Yki[act] hemolymph, the excitation response of NPF-P1 neurons was significantly repressed (Fig. 2h, Supplementary Fig. 2e, and Supplementary Movie 2). Taken together, our data suggest that tumor-derived secreted factors in the hemolymph of Yki[act] male flies silence NPF neurons. This subsequently lowers *NPF* expression in the brain, making flies

nonresponsive to internal nutritional needs, which leads to anorexia prior to organ wasting.

## ImpL2 and upd3 together cause NPF-linked anorexia in Yki[act] males

We next searched for tumor-derived secreted factors responsible for promoting anorexia. A previous study testing eye-tumor-derived anorexia in flies reported that elevated expression of insulin-like peptide dilp8 was responsible for reduced food intake[44]. We did not observe an increase of *dilp8* expression in Yki[act] male flies (Supplementary Fig. 3a). Also, conditional knockdown of *dilp8* in the gut (*elavGal80; esg[TS] > Yki[act] + dilp8[RNAi]*) of Yki[act] male flies by d6 did not increase food intake (Supplementary Fig. 3b). Together, these data suggest that tumor-derived dilp8 does not regulate the anorexic behavior of Yki[act] male flies.

We reasoned that if the onset of anorexia is linked to organ wasting, then tumor-derived secreted factors that regulate organ wasting in Yki[act] male flies also regulate NPF-linked anorexia. The conserved inflammatory cytokine upd3 and the insulin antagonist ImpL2 are two factors that are secreted in excess during gut Yki-tumor growth and regulate organ wasting[9–11,14]. In addition, insulin signaling (from insulin-producing cells in the brain) and upd-related signaling from the brain and fat body are part of the satiety-promoting network in *Drosophila* that stops feeding and represses orexigenic signaling, including NPF[12,28,34,52]. After testing the levels of *upd3* and *ImpL2* expression in Yki[act] male flies, we observed that by d4, they are both significantly increased in the whole body (Fig. 3a) and gut (Supplementary Fig. 3c), aligning with the early decrease of *NPF* levels by d4 (Fig. 2b).

We next asked if either or both of these tumor-secreted cytokines are responsible for anorexia prior to organ wasting. Conditional knockdown of *upd3* in the gut (*elavGal80; esg[TS] > Yki[act] + upd3[RNAi]*) increased food intake in Yki[act] male flies at d6 (Supplementary Fig. 3b). However, these data are difficult to interpret, since upd3 drives gut tumor growth[14] and therefore a rescue in feeding behavior is due to inhibition of tumor development and not a direct impact on feeding-related signaling. Conditional knockdown of *ImpL2* in the gut of Yki[act] male flies (*elavGal80; esg[TS] > Yki[act] + ImpL2[RNAi]*) significantly increased feeding behavior at d5 and d6 (Fig. 3b, c) and restored *NPF* expression to levels similar to control (Fig. 3d). Since ImpL2 does not impact tumor growth[9,19] these data support that ImpL2 signaling from the gut tumor promotes the reduction of *NPF* and subsequent anorexia in Yki[act] male flies.

Previous work tested the role of ImpL2 in organ wasting independent of tumor growth by ectopically over-expressing *ImpL2* in large tissues, such as the muscle, and observed that excessive circulation of ImpL2 in tumor-free flies is sufficient to promote organ wasting[19]. To test for a role of ImpL2 in anorexia independent of gut tumor growth, we took a similar approach and conditionally overexpressed *ImpL2* from a large tissue. We chose the muscle because it was used previously[19] and because in gut Yki-tumor flies elevated amounts of circulating ImpL2 originate primarily from the gut but have also been reported to come from the muscle[14]. We therefore overexpressed *ImpL2* from the muscle of male flies for 4 days using the *mef2-Gal4* driver together with the *Tubulin*-Gal80[TS] repressor (referred as *mef2[TS]*). *Mef2[TS] > ImpL2* flies did not exhibit an anorexic phenotype (Fig. 3e, f), in agreement with a previous report[19]. However, *NPF* was slightly reduced (~20% decrease) in *mef2[TS] > ImpL2* male flies compared to control (Fig. 3g). These data suggest that increase of circulating ImpL2 moderately lowers *NPF*, but the impact is not sufficient to reduce food intake.

Since knocking down *ImpL2* in Yki[act] male flies restored both *NPF* and feeding behavior (Fig. 3b–d), we speculated that ImpL2 may work in synergy with another tumor-secreted factor. Overexpression of *upd3* from the muscle (*mef2[TS] > upd3*) did not reduce food intake

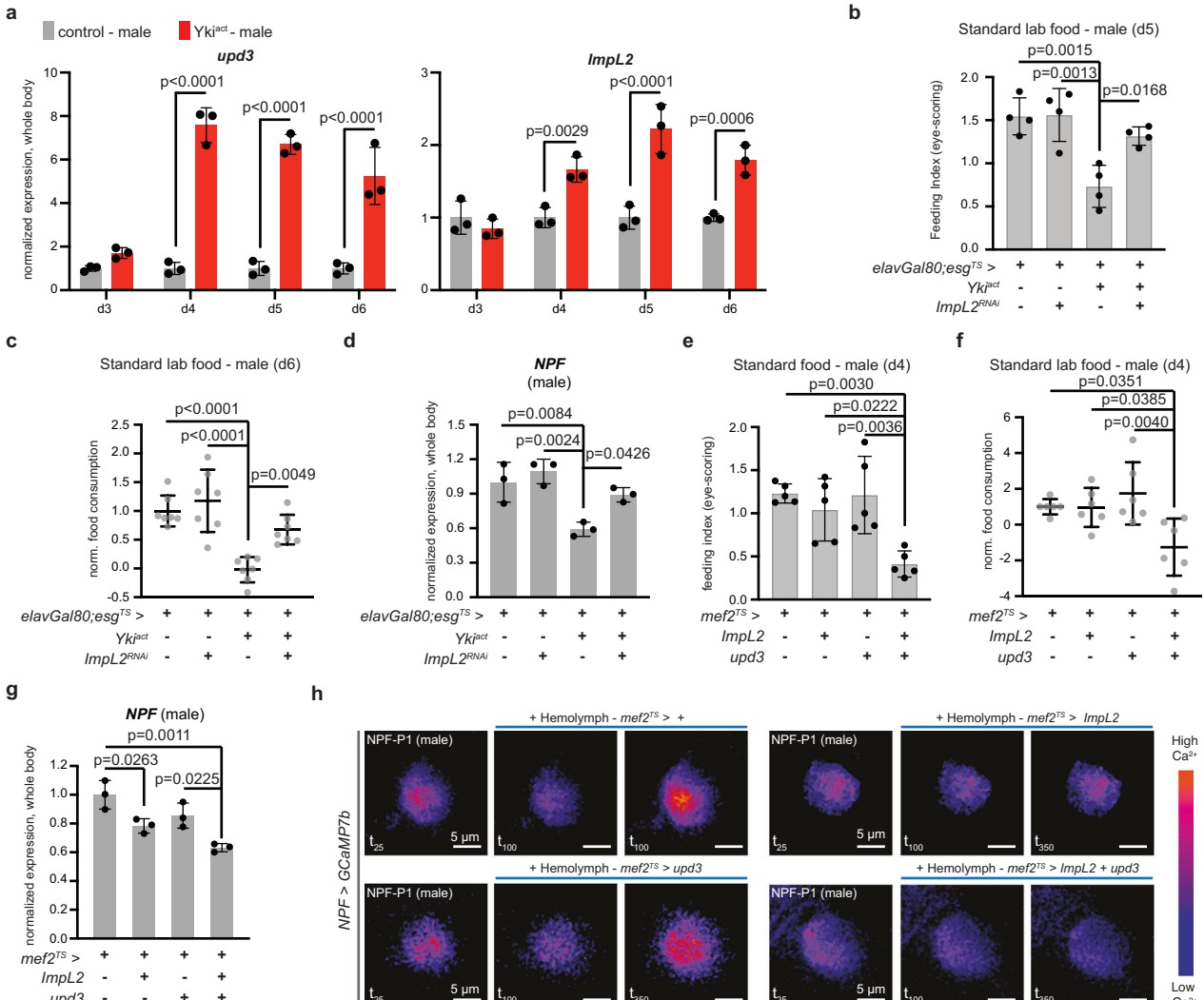

**Fig. 3 | Combined ImpL2 and upd3 increase leads to anorexic behavior in males.**
**a** Expression levels of *ImpL2* and *upd3* from the whole body of male control (gray) and Yki[act] (red) on d3, d4, d5, and d6. *n* = 3 biological replicates per genotype, timepoint. Three independent experiments. Normalized to control. Two-way Anova (Sidak's test). **b** Eye-scoring assay. Control, Yki[act], *elavGal80;esg[TS] > ImpL2[RNAi]* (*ImpL2* knocked down in the gut), *elavGal80;esg[TS] > yki[act] + ImpL2[RNAi]* (Yki[act] flies with *ImpL2* knocked down) male flies on d5. *n* = 4 biological replicates per genotype. Three independent experiments. One-way Anova (Tukey's test). **c** Blue-dye (spectrophotometry) assay. Genotypes like (**b**) on d6. Normalized to control. *n* = 7 biological replicates per genotype. Three independent experiments. One-way Anova (Tukey's test). **d** *NPF* levels from the whole body of d5 males. Genotypes like (**b**). *n* = 3 biological replicates per genotype. Three independent experiments. Normalized to control. One-way Anova (Tukey's test). **e** Eye-scoring assay. Control

(*mef2[TS] > +*), flies with *Impl2* (*mef2[TS] > ImpL2*), or with *upd3* (*mef2[TS] > upd3*), or *Impl2* and *upd3* (*mef2[TS] > ImpL2 + upd3*) overexpressed from the muscle for 4 days. *n* = 5 biological replicates per genotype. 3 independent experiments. One-way Anova (Tukey's test). **f** Blue-dye (spectrophotometry) assay. Genotypes like (**e**). Normalized to control. *n* = 6 biological replicates per genotype. Four independent experiments. One-way Anova (Tukey's test). **g** *NPF* from the whole body of d4 males. Genotypes like (**e**). *n* = 3 biological replicates per genotype. Normalized to control. One-way Anova (Tukey's test). Three independent experiments. **h** Color-coded sequential frames of NPF-P1 neurons from *NPF > GCAMP7b* male brains before (*t₂₅*) and after (*t₁₀₀*), (*t₃₅₀*) hemolymph addition from *mef2[TS] > +*, *mef2[TS] > ImpL2*, *mef2[TS] > upd3*, *mef2[TS] > ImpL2 + upd3* male flies. Scale bar: 5 μm. Supplementary Movies 3–6. Individual graphs: Supplementary Fig. 3g–j. Mean ± SD. Exact *p*-values are shown. Source data are provided as Source data file.

(Fig. 3e, f) and did not decrease *NPF* expression (Fig. 3g). However, the combined overexpression of *ImpL2* and *upd3* (*mef2[TS] > ImpL2 + upd3*) was sufficient to significantly reduce food intake (Fig. 3e, f), reminiscent of gut Yki-induced anorexia. In addition, joint increase of *ImpL2* and *upd3* from the muscle caused a significant ~40% decrease of *NPF* in the whole body and head (Fig. 3g and Supplementary Fig. 3d), phenocopying Yki[act] male flies. Together, these data suggest that ImpL2 and upd3 work synergistically to cause a large decrease in *NPF* levels that is sufficient to drive anorexia.

One of the reported functions of upd3 and IL-6 in fly and mouse tumor models, respectively, is to disrupt the BBB[20] and increase the passage of factors to the brain. We speculated that during gut Yki-tumor growth, upd3 aids ImpL2 by allowing aberrant circulation to the

brain. To test this, we knocked down the *JAK Drosophila* homolog *hopscotch* (*hop*) with the BBB-driver *repo-Gal4* (together with the *Tubulin*-Gal80[TS] repressor, referred as *repo[TS]*), thus preventing upd3 from activating the JAK/STAT pathway and possibly impairing BBB permeability[20]. Simultaneously, we induced gut Yki-tumor growth by expressing the Yki[act] transgene (*LexAop-Yki[act]*) with the intestinal progenitor LexA driver *esg-LexA:GAD* (repressed by the neuronal *elav*-Gal80 and *Tubulin*-Gal80[TS] repressor, referred as *elavGal80; esgLexA:GAD[TS]>*). We found that conditional knockdown of *hop* in the BBB during gut Yki-tumor growth (*elavGal80; esgLexA:GAD[TS] > LexAopYki[act] + repo[TS] > hop[RNAi]*) significantly increased food intake by d5 (Supplementary Fig. 3e), supporting that aberrant passage to the brain due to upd3 affecting the BBB is required for

anorexia prior to organ wasting. *NPF* expression was subtly increased but not significantly restored when *hop* was knocked down in the BBB during Yki-tumor growth (Supplementary Fig. 3f). These flies (*elav-Gal80; esgLexA:GAD^TS^ > LexAopYki^act^ + repo^TS^ > hop^RNAi^*) exhibited a ~20% reduction of *NPF* compared to control (Supplementary Fig. 3f), which we speculate is due to remaining ImpL2 signaling that moderately lowers *NPF*.

To further test the impact of unrestricted ImpL2 and upd3 signaling in NPF neurons, we performed an ex vivo brain assay, using hemolymph from male flies with *ImpL2* or *upd3* or both over-expressed from the muscle for 4 days (Fig. 3h, Supplementary Fig. 3g–j, and Supplementary Movie 3–6). Excitation of NPF-P1 neurons was not inhibited when we added hemolymph from *mef2^TS^ > upd3* flies (Fig. 3h, Supplementary Fig. 3g, j, and Supplementary Movie 4), supporting that high levels of upd3 alone does not directly impact NPF signaling. However, adding hemolymph from *mef2^TS^ > ImpL2* flies was sufficient to prevent the excitation of NPF-P1 neurons (Fig. 3h, Supplementary Fig. 3h, j, and Supplementary Movie 5). Similarly, the addition of hemolymph from *mef2^TS^ > upd3 + ImpL2* flies, significantly prevented the excitation of NPF-P1 neurons (Fig. 3h, Supplementary Fig. 3i, j, and Supplementary Movie 6), resembling the addition of Yki^act^ hemolymph.

Altogether, our data suggest that excessive increase of ImpL2 drives the reduction of neuronal NPF signaling and that upd3 augments the inhibitory impact of ImpL2, leading to anorexia. We propose that upd3 affects the BBB and increases the passage of ImpL2 to the brain. Given the multifaceted role of upd3 during tumor growth[11,14], we cannot exclude that upd3 signaling impacts additional tissues, together with the BBB, to augment the function of ImpL2. Altogether, our findings support that the early synergistic function of tumor-induced ImpL2 and upd3 is required for strong reduction of NPF signaling from the brain and subsequent anorexia in Yki^act^ male flies.

### Low neuronal NPF drives protein-appetite reduction in Yki^act^ males

CC patients often experience difficulty ingesting amino acid-rich foods, such as meat[4]. Interestingly, when we tested food rich for different macronutrients, we found that Yki^act^ male flies on d6 (onset of organ wasting) exhibit anorexic behavior for yeast (protein-rich food) and essential amino acids (EAA), whereas food intake for sugar-rich, fat-rich food and nonessential amino acids (NEAA) was similar to control (Fig. 4a). These data support that reduced food intake in Yki^act^ male flies is specific to protein-rich food.

Nutrient-specific appetites drive the choice for foods with precise macronutrients so that nutritional homeostasis and optimal fitness is maintained—a behavior conserved among animals, including flies[53,54]. We therefore asked whether the observed protein malnutrition is the consequence of an earlier imbalance in macronutrient-specific food choices, occurring before the onset of anorexia. To test this, we used the FlyPAD to conduct a two-choice behavioral assay for the two main macronutrients in the fly's diet, protein and sugar (yeast vs. sugar preference assay)[33]. When testing control male flies, we observed that their choice for protein-rich food (yeast) over sugar was elevated at dusk compared to dawn (Supplementary Fig. 4a), indicating that macronutrient preference is likely regulated by daily rhythms, similar to overall food intake[55]. Taking advantage of this observation, we assayed Yki^act^ male flies at dusk on d4 (one day prior to anorexia) and discovered that they choose significantly less protein-rich food compared to control (Fig. 4b). Day 4 of gut Yki-tumor growth in male flies is also when *upd3* and *ImpL2* are significantly elevated, and *NPF* is reduced (Figs. 2b and 3a).

Next, we tested the synergistic impact of upd3 and ImpL2 in macronutrient-specific intake and appetite and found that *mef2^TS^ > upd3 + ImpL2* male flies phenocopy the characteristics of Yki^act^ malnutrition (Fig. 4c, d). Specifically, the anorexic

*mef2^TS^ > upd3 + ImpL2* male flies eat significantly less protein-rich and EAA-rich food, whereas intake of sugar-rich, fat-rich and NEAA-rich food remains similar to control (Fig. 4c). In addition, *mef2^TS^ > upd3 + ImpL2* flies choose significantly less protein-rich food (Fig. 4d) on d1 (a timepoint prior to their anorexic behavior, Supplementary Fig. 4b, c), reminiscent of Yki^act^ male flies.

We also tested whether the early imbalance in protein-appetite is linked to NPF signaling. Systemic TrpA1-induced increase of NPF signaling in Yki^act^ male flies by d4 (*elavGal80; esg^TS^ > yki^act^ + NPFLexA > LexAopTrpA1*) was sufficient to rescue the low preference for protein-rich food (Fig. 4e). In addition, TrpA1-induced increase of NPF signaling only from the gut (*elavGal80;esg^TS^ > yki^act^ + NPFLexA:GAD > LexAopTrpA1*) did not rescue the low preference for protein and further reduced the choice for sugar (Fig. 4e). Taken together, we propose that by d4 of tumor induction (2 days prior to organ wasting), excess of ImpL2 and upd3 reduces neuronal NPF signaling, triggering a decline in protein appetite and making Yki^act^ male flies prone to protein malnutrition.

### NPF-linked malnutrition reduces weight and survival of Yki^act^ males

Protein malnutrition and subsequent internal amino acid (AA) scarcity worsens tissue breakdown in CC patients and is associated with poor prognosis[6,56]. We therefore tested the impact of NPF-linked protein malnutrition in organ wasting symptoms, such as increased lipolysis and weight loss. TrpA1-induced increase of NPF signaling in Yki^act^ male flies (*elavGal80; esg^TS^ > yki^act^ + NPFLexA > LexAopTrpA1*) did not rescue the reduction in triglycerides by d6 (Supplementary Fig. 5a). This indicates that NPF decline does not impact lipolysis, and this organ wasting symptom is regulated by distinct pathways unrelated to feeding. However, we observed that TrpA1-induced increase of NPF signaling in Yki^act^ male flies restored the dry weight to levels similar to control at d6 (Fig. 5a). In addition, an increase in NPF signaling solely from the gut did not rescue the dry weight in Yki^act^ male flies (Supplementary Fig. 5b). Together, these data support that early reduction in neuronal NPF signaling contributes to the onset of weight loss by d6 in Yki^act^ male flies.

Further supporting that weight loss is linked to significant decrease in *NPF* and ensuing changes in feeding behavior, joint *ImpL2* and *upd3* overexpression from the muscle for 4 days (*mef2^TS^ > upd3 + ImpL2*) led to a significant reduction in dry weight (Fig. 5b). By contrast, *mef2^TS^ > upd3* and *mef2^TS^ > ImpL2* flies did not exhibit a significant reduction in dry weight (Fig. 5b).

We reasoned that if organ-wasting symptoms like weight loss are linked to prior unmet nutritional needs, then an early increase of NPF signaling should improve the overall outcome of organ wasting. Indeed, *elavGal80; esg^TS^ > yki^act^ + NPFLexA > LexAopTrpA1* male flies had a significantly higher probability of survival compared to Yki^act^ flies (Fig. 5c). Specifically, increased NPF signaling caused Yki-tumor flies to extend their median lifespan (the day survival curve passes 50% survival point) by 5 days and their maximum lifespan (0% survival probability) was similar to control (Fig. 5c and Supplementary Fig. 5c). In addition, we tested the probability of death by d21/24 (the approximate half-life of control flies which is ~23 days) and observed a significant decrease in *elavGal80; esg^TS^ > yki^act^ + NPFLexA > LexAopTrpA1* flies compared to Yki^act^ flies (Fig. 5d). Therefore, our findings support that early rescue of NPF signaling and fulfillment of nutritional needs prior to organ wasting, averts the onset of weight loss and mitigates the high risk of death in Yki^act^ male flies.

Altogether, we propose that the joint excessive increase of upd3 and ImpL2 by d4 in gut Yki-tumor male flies leads to reduction of neuronal NPF signaling, causing low preference for protein-rich food and anorexia prior to organ wasting. This early nutritional impairment drives aspects of organ wasting, like weight loss and increases the probability of death, all of which are rescued

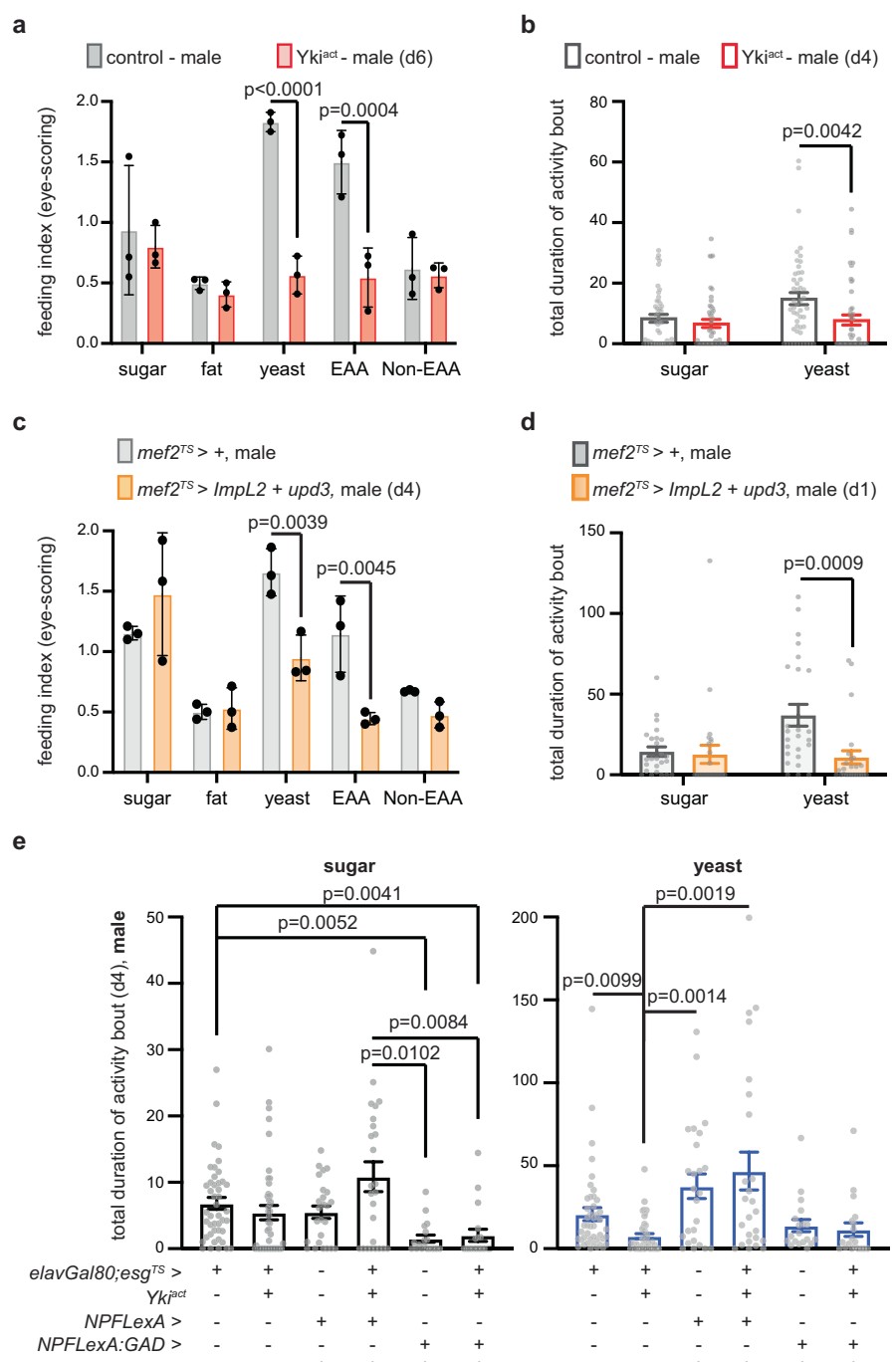

**Fig. 4 | NPF activation in male Yki^act flies rescues protein-rich food preferences.**
**a** Eye-scoring assay with different diets on d6. Control (gray) and Yki^act (red). $n = 3$ biological replicates per genotype and diet. Three independent experiments. Two-way Anova (Sidak's test). yeast: protein-rich food. EAA: food rich with essential amino acids. Non-EAA: food rich with nonessential amino acids. **b** FlyPAD two choice-assay. Genotypes like (**a**) on d4 dusk. $n = 48$ flies per genotype. Kruskal–Wallis Dunn's test. Three independent experiments. **c** Eye-scoring assay with different diets. Control (*mef2^TS > +*, gray) and male flies with *ImpL2* and *upd3* overexpressed from the muscle for 4 days (*mef2^TS > ImpL2 + upd3*, orange). $n = 3$ biological replicates per genotype and diet. Two-way Anova (Sidak's test). Three independent experiments. **d** FlyPAD two choice-assay. Genotypes like (**c**), d1 (-18 h) at dusk. $n = 25$ flies per genotype.

Kruskal–Wallis Dunn's test. Four independent experiments. **e** FlyPAD two choice-assay on d4 dusk. Sugar: black; yeast: blue. Male control (*elavGal80;esg^TS > +*, $n = 44$ flies), Yki^act (*elavGal80;esg^TS yki^act*, $n = 44$ flies), flies with increased NPF signaling (*NPFLexA > LexAopTrpA1 + elavGal80;esg^TS > +*, $n = 25$ flies), flies with increased NPF signaling and Yki gut-tumor (*NPFLexA > LexAopTrpA1 + elavGal80;esg^TS > yki^act*, $n = 25$ flies), increased gut NPF signaling (*NPFLexA:GAD > LexAopTrpA1 + elavGal80;esg^TS > +*, $n = 19$ flies), increased gut NPF signaling and Yki gut-tumor (*NPFLexA:GAD > LexAopTrpA1 + elavGal80;esg^TS > yki^act*, $n = 19$ flies). Six independent experiments. Kruskal–Wallis Dunn's test. Activity bout: seconds. Mean ± SD (**a**, **c**), SEM (**b**, **d**, **e**). Exact *p*-values are shown. Source data are provided as Source data file.

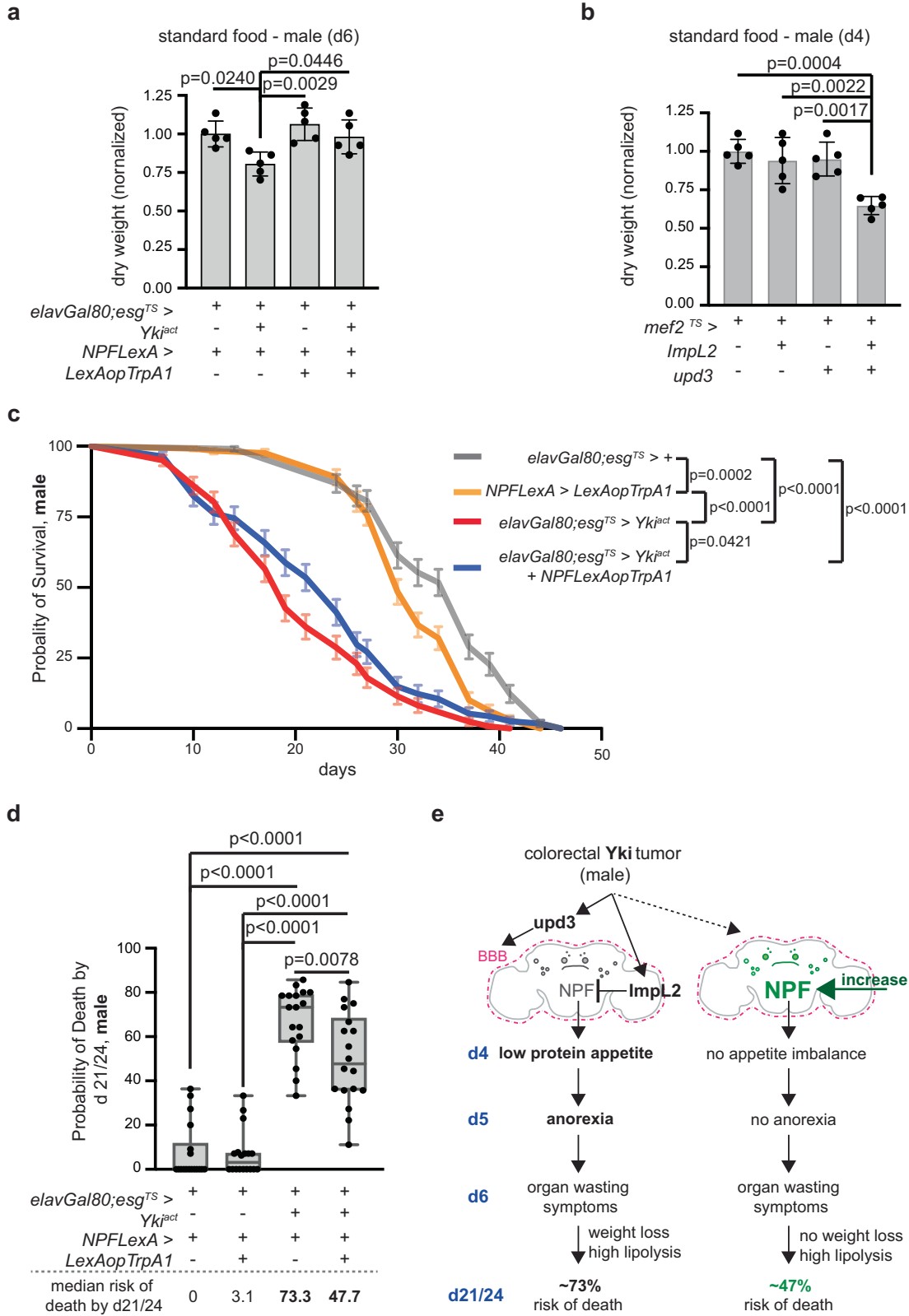

once neuronal NPF signaling is restored prior to organ wasting (Fig. 5e).

## Discussion

Malnutrition in CC is associated with poor prognosis and high death risk[6]. Here, we provide evidence that early rescue of the conserved orexigenic NPF/NPY signaling in gut Yki-tumor flies restores nutrient-specific appetites and food intake prior to organ wasting, subsequently preventing weight loss and reducing the high death risk during organ wasting. There are conflicting reports about the role and impact of NPY in CC[3,57,58]. Our findings support a re-evaluation for the role of NPY signaling, particularly at the early pre-cachexia stages.

Moreover, we discovered that an early decrease in neuronal NPF signaling drives low protein-appetites prior to anorexia and organ

**Fig. 5 | NPF activation in male Yki$^{act}$ flies rescues weight loss and increases survival. a** Dry weight assay. Male control (*elavGal80;esg$^{TS}$/+; NPFLexA/+*), Yki$^{act}$ (*elavGal80;esg$^{TS}$ > yki$^{act}$ + NPFLexA > +*), flies with increased NPF signaling (*NPFLexA > LexAopTrpA1 + elavGal80;esg$^{TS}$ > +*) and flies with increased NPF signaling and Yki gut-tumor (*NPFLexA > LexAopTrpA1 + elavGal80;esg$^{TS}$ > yki$^{act}$*) on d6. *n* = 5 biological replicates per genotype. Three independent experiments. Normalized to control. One-way Anova (Tukey's test). **b** Dry weight assay. Control (*mef2$^{TS}$ > +*), flies with ImpL2 (*mef2$^{TS}$ > ImpL2*), upd3 (*mef2$^{TS}$ > upd3*), ImpL2 and upd3 (*mef2$^{TS}$ > ImpL2 + upd3*) overexpressed from the muscle for 4 days. *n* = 5 biological replicates per genotype. Three independent experiments. Normalized to control. One-way Anova (Tukey's test). **c** Probability of survival of male control (*elavGal80;esg$^{TS}$ > +*, gray, *n* = 114 flies out of 9 biological replicates), Yki$^{act}$ (*elavGal80;esg$^{TS}$ > yki$^{act}$*, red, *n* = 122 flies out of 9 biological replicates), flies with increased NPF signaling (*NPFLexA > LexAopTrpA1*, orange, *n* = 128 flies out of 9 biological replicates), and flies with increased NPF signaling and Yki gut-tumor (*elavGal80;esg$^{TS}$ > yki$^{act}$ + NPFLexA > LexAopTrpA1*, blue, *n* = 114 flies out of 9

biological replicates). Log-rank (Mantel–Cox) test. Nine independent experiments. **d** Probability of death for the 9 biological replicates on d21 and 24 per genotype. Genotypes like (**c**). Dotted line (below): median risk of death. *n* = 18 biological replicates per genotype. One-way Anova (Tukey's test). Nine independent experiments. **e** Model: by d4 of Yki-tumor growth, ImpL2 silences NPF neurons, and upd3 amplifies this effect by impacting the blood-brain barrier (BBB). Strong reduction of NPF in the brain due to the synergistic effect of ImpL2 and upd3 reduces protein-rich food preference at d4 and causes anorexia at d5. This protein malnutrition initiates weight loss and worsens the risk of death. When Yki$^{act}$ males meet their nutritional needs due to early increase of NPF signaling in the brain, the onset of weight loss by d6 is prevented, and the risk of death by d21/24 is reduced to ~47%. Mean ± SD (**a**, **b**). Survival curve: mean ± SEM (**c**). Boxplots (**d**): minimum, 25th percentile, median, 75th percentile, and maximum values. Figure 5e was drawn with Adobe Illustrator. Exact *p*-values are shown. Source data are provided as Source data file.

wasting. Our work underscores the idea that a comprehensive characterization of nutrient-specific appetites during pre-cachexia stages could be beneficial to the emerging demand for early diagnostic indicators, and pre-cachexia multimodal interventions to extend and improve the quality of life for CC patients[8]. Specifically, our findings support that the tumor-compromised animal is prone to amino acid malnutrition prior to the first signs of cachexia, thus worsening the overall outcome of organ wasting. These data align with reports of CC patients being carnitine deficient (deficient for lysine and methionine)[59] and reports that this deficiency contributes to progression of cancer to cancer cachexia[60–62]. However, the underlying mechanism remains unclear. It would be interesting to use the *Drosophila* gut Yki-tumor model to decipher the exact EAA responsible for low protein-appetite prior to organ wasting and examine how this specific early EAA malnutrition impacts the advent of organ wasting.

Reduced insulin signaling and elevated inflammation are intertwined, with recent work reporting that these two mechanisms could be indicators of poor prognosis in CC patients[63]. In addition, brain inflammation and BBB impairment is associated with CC[64], but the underlying mechanism remains unclear. We unraveled a sequence of early tumor-induced events involving reduced insulin signaling, elevated inflammation and potential BBB impairment that cause NPF-linked anorexia prior to organ wasting. Our findings suggest that excess of the conserved inflammatory upd3 alone does not directly impact NPF signaling in gut Yki-tumor flies, despite reports that some NPF neurons express the upd-receptor[28]. Instead, we propose that upd3 signaling indirectly regulates anorexia prior to organ wasting by impairing the BBB. Moreover, NPF and insulin signaling are part of a complex feedback loop that regulates feeding in flies[12]. ImpL2 sequesters insulin-like peptides[65] and is also reported to ensure increased precision of local insulin signaling in the larval brain[66]. We found that excess of the conserved insulin antagonist ImpL2 prevents the activation of NPF neurons and is required for NPF-linked anorexia in gut Yki-tumor flies. Although increase of ImpL2 alone does not cause anorexia, our findings support that increase of ImpL2 in the presence of upd3 strongly decrease *NPF* to levels sufficient to cause anorexia. Taken together, we propose that ImpL2 reduces the activation of NPF neurons while upd3 augments the role of ImpL2 by disrupting the BBB and allowing aberrant passage of ImpL2 into the brain. The ensuing strong decrease of neuronal NPF signaling causes anorexia and exacerbates the risk of death during organ wasting.

In humans, CC exhibits sex differences, with male patients having a higher prevalence[67]. While sexual dimorphism in pathways associated with CC has been reported, the reason for the difference in response to CC between male and female patients remains unclear[68].

The *Drosophila* gut Yki-tumor model is an attractive model to address this knowledge gap, and our findings support that feeding-related signals prior to organ wasting exhibit differences in male and female flies. For example, it would be interesting to decipher why the main conserved satiety regulator upd2/leptin is significantly increased specifically in females prior to organ wasting.

Overall, this work advances our current knowledge of events occurring prior to organ wasting by linking tumor-induced: (i) inflammation, blood brain barrier impairment and reduction of insulin signaling to the advent of anorexia prior to organ wasting; (ii) NPF dysfunction to early weight loss and high death risk; (iii) decrease of protein-specific appetites to the onset of organ wasting.

## Methods

### *Drosophila* stocks, husbandry and conditions

Flies were raised at ~18 °C on Perrimon lab food (corn meal/ agar medium, recipe below). All crosses were maintained at ~18 °C, and tumor induction (or/and neuronal activation) was conducted by transferring flies from ~18 to 29 °C. Experiments were conducted with mated female and male flies, 3–5 days old. This study primarily focuses on male flies, so all main Figures include data from male flies. Data from female flies are on Supplementary Figs. 1a–m and 2a all other Supplementary Figs. include data from male flies. All experiments were conducted with flies fed *ad libitum,* and flies were trained in 12:12 (light: dark) cycles. Detailed genotypes per figure are in the Supplementary Information. The raw data generated in this study are provided in the Source data file.

### Fly stocks

The fly lines used in this study have been described previously and include: *NPF-LexA* (BDSC: 83720), *NPF-LexA:GAD* (*BDSC: 84422*), *NPF-Gal4* (BDSC: 25681), *Tubulin-Gal80$^{TS}$* (BDSC:7018,7019), *UAS-Yki$^{3SA}$* (BDSC: 28817), *UAS-emptyVK37* (gift from H. Bellen), *UAS-ImpL2$^{RNAi}$* (NIG:15009R3), *UAS-upd3$^{RNAi}$* (BDSC:32859), *UAS-dilp8$^{RNAi}$* (BDSC:80436), *UAS-ImpL2$^{69}$, UAS-upd3$^{70}$, UAS-APC$^{RNAi}$* (BDSC: 28582), *UAS-Raf$^{F179}$* (*UAS-Raf$^{GOF}$*, BDSC: 2033), *UAS-Luciferase$^{RNAi}$* (BDSC: 31603), *13x-LexAop2-IVS-TrpA1* (*lexAopTrpA1*, gift from G. Rubin), *20XUAS-6XmCherry-HA* (BDSC: 52268), *20XUAS-IVS-jGCaMP7b* (*UAS-GCAMP7b*, BDSC: 79029), *UAS-nuclearGFP* (*UAS-unc84-GFP*), *LexAop-Yki3SA-GFP* (*LexAopYki$^{act}$*)[10], *UAS-hop$^{RNAi}$* (BDSC: 32966); Perrimon lab stocks: *elav-Gal80, esg-Gal4 Tubulin-Gal80$^{TS}$ UAS-GFP* (*esg$^{TS}$*), *dmef2-GAL4* (*mef2*), *esg-LexA:GAD* (BDSC: 66332), *repo-Gal4* (BDSC: 7415).

### Fly food recipes

*Perrimon lab fly food (standard lab food)*: 12.7 g/liter deactivated yeast, 7.3 g/liter soy flour, 53.5 g/liter cornmeal, 0.4% agar, 4.2 g/liter malt, 5.6% corn syrup, 0.3% propionic acid, and 1% Tegosept/ ethanol.

During the experiments, flies were flipped in new vials with standard lab food every other day. *Food with antibiotics*: antibiotics were added in the lab food at a final concentration of 1 mg/ml kanamycin (Kan, Sigma, K1377), 640 μg/ml ampicillin (VWR 45000-614) and 640 μg/ml doxycycline (Dox, Thermo Scientific J60579.22). Flies were flipped in fresh vials of food with antibiotics every day. *Food with molasses*: we used a pre-made molasses formula as described by Genesee Scientific (66–116). Flies were flipped in new vials with molasses food every other day. *Sugar-rich food (sugar)*: we used 5% sucrose (Fisher, BP220-10) in water for blue dye and FlyPAD assays. *Fat-rich food (fat)*: we used 20% coconut oil mixed in sugar-rich food (5% sucrose in water). Coconut oil was warmed at 42 °C for 2 h prior to mixing with sugar-rich food. *Protein-rich food (yeast)*: for blue dye assays, we used 5% inactivated dry yeast (Genesee Scientific, 62–107) in water. For FlyPAD, we used 10% inactivated yeast in water. *Amino acid (AA)-rich food*: for essential AA (EAA): 8 g of L-arginine (Sigma, A5006), 10 g of L-histidine (Sigma, H6034), 19 g of L-lysine (Sigma, P6516), 8 g of L-methionine (Sigma, M5308), 13 g of L-phenylalanine (Sigma, P2126), 20 g of L-threonine (Sigma, T8625), 5 g of L-tryptophan (Sigma, T8941), and 28 g of L-valine (Sigma, V0513), were all mixed in 1 L water. 140.45 ml of this EAA stock solution was then added in 1 L of water, together with 4.21 g L-isoleucine (Sigma, I7403), 2.81 g L-leucine (Sigma, L8912) and 17.12 g sucrose (Fisher, BP220-10), to make the final EAA solution that was used in feeding assays. For non-EAA: 35 g of L-alanine (Sigma, A7469), 17 g of L-asparagine (Sigma, A0884), 17 g of aspartic acid (Sigma, A7219), 0.5 g of L-cysteine (Sigma, C6852), 25 g of L-glutamine (Sigma, G8540), 32 g of Glycine (Sigma, G8790), 15 g of L-proline (Sigma, P5607), and 19 g of L-serine (Sigma, S4311) were all mixed in 1 L water. 97.52 ml of this non-EAA stock solution was then added in 1 L of water, together with 24.38 ml of L-glutamate (Sigma, G8415), 1.61 g of L-tyrosine (Sigma, T8566), 17.12 g sucrose (Fisher, BP220-10), to make the final non-EAA solution that was used in feeding assays.

## Feeding assays

All flies were trained in 12:12 (light: dark) cycles at 29 °C (unless flies were assayed on d0, where they remained at 18 °C), before performing behavioral experiments.

FlyPAD: to assay for standard lab food, we added a droplet (3 μl) of standard lab food that contained 1% agarose (VWR, 490001-584) in one of the two wells found in each behavioral arena. The other well was covered with tape. We monitored the feeding activity (total duration of activity bout, seconds)[33] of individual flies (from the well with the lab food) for 1 h at RT between ZT0-ZT2 (12:12 LD). For the two-choice assay, male flies were given the choice between droplets (3 μl) containing 1% agarose with either 5% sucrose (sugar, Fisher, BP220-10) or 10% inactivated yeast (protein-rich, Genesee Scientific, 62–107). For the two-choice assay, flies were assayed for 30 min at RT between ZT10-ZT12 (12:12LD). For each assay, flies were placed into the behavioral arenas where they were left to acclimate for five to six minutes before data acquisition. Data acquisition and processing (Bonsai data-stream processing program) was done using the developer's Flypad software (http://www.flypad.pt)[33]. All FlyPAD assays were conducted with flies fed *ad libitum* standard lab prior to the assay. After each FlyPAD assay, we checked the state of food and flies in each behavioral arena. If by the end of an assay the food had completely dried out or a fly was injured or dead or had escaped the arena, then these experiments were discarded.

Blue dye: flies were placed for 2 h in vials that contained food mixed with blue dye, before measuring food intake using eye-scoring or color spectrophotometry. All experiments were done at 29 °C between ZT0-ZT2 (12:12 LD). Each biological replicate per genotype consisted of 20 flies for spectrophotometry and of ~15–20 flies for eye-scoring. For the eye-scoring assay in Supplementary Fig. 3e -10–15 flies per biological replicate per genotype were used. For eye-scoring, flies with no blue dye in their abdomen (no blue dye on crop and gut) were

scored with 0, flies with blue dye in more than half of the abdomen (blue dye on both crop and gut) were scored with 2, and flies with blue dye in less than half of the abdomen (either crop or gut) were scored with 1. For color spectrophotometry, we homogenized each sample in 750 μl 1% Triton-X-100 (Sigma X100) using 1 mm beads in a bullet blender homogenizer (Next Advance). Homogenates were then centrifuged and filtered with 40 μm cell strainer into fresh Eppendorf tubes. One hundred fifty micrometers per homogenate were added in microplates that were read at 630nm with the SpectraMax Paradigm Multi-mode microplate reader (Molecular Devices). For each different genotype, we included a blank sample (no blue dye in the food, 20 flies per blank sample) that was treated identically to account for background differences. Food consumption was calculated by subtracting the value of the blank sample per genotype. To calculate the fold-change, the value of food consumption for each genotype (after the blank was subtracted) was normalized to the control genotype.

Blue dye food was prepared the day of the assay from a stock solution of 0.125 g/ml blue dye (FD&C blue dye #1, VWR, 700010-048) that was kept in the dark at 4 °C and was remade monthly. To assay standard lab food or food with molasses, 8 μl/ml of blue dye stock solution was mixed with warmed food and then added in empty vials that were left to solidify prior to usage. For standard lab food with antibiotics, 8 μl/ml of blue dye stock solution were first mixed with warmed lab food, and then antibiotics were added before the food solidified in the empty vials. For blue dye assays with food rich for different macronutrients (sugar-, protein-, fat-, AA- and EAA- rich food), we mixed 4 μl/ml of blue dye stock with each food and then added 1 ml of the mixture in individual fly vials that had in the bottom a Kimwipe cut in half (KimTech Science) to absorb the mixture. All blue dye feeding assays were conducted with flies fed standard lab *ad libitum* prior to each experiment, except for the experiments conducted with antibiotics or molasses, where flies were fed *ad libitum* food with antibiotics or molasses, respectively.

## Immunohistochemistry and imaging

Brains were dissected and fixed in 4% PFA in PBS for 30 min at RT. They were washed with 1% Triton X-100 (Sigma X100) in PBS, incubated for 1 h at RT in blocking solution (for brains see below) and then stained with primary antibody in blocking solution overnight at 4 °C. Next, brains were washed with 0.25% Triton X-100 in PBS and then stained with secondary antibody and DAPI in blocking solution at 4 °C overnight in the dark. Finally, samples were washed with 0.25% Triton X-100 in PBS and mounted with Vectashield. Similarly, guts were dissected and fixed in 4% PFA, washed in PBS and incubated for 1 h in blocking solution (for guts). Next, they were stained with primary antibody overnight at 4 °C, then washed in 0.1% Triton X-100 in PBS and stained with secondary antibody and DAPI in blocking solution at 4 °C overnight in the dark. Guts were washed with 0.1% Triton X-100 in PBS before being mounted in Vectashield. Blocking solution for brains: 10% normal donkey serum, 0.25% Triton X-100 in PBS. Blocking solution for guts: 10% normal donkey serum, 0.1% Triton X-100 in PBS. Primary antibodies: rabbit anti-NPF (RayBiotech, RB-19-001-20, 1:500), rabbit anti-GFP (Invitrogen, A6455; 1:3000), mouse anti-GFP (Invitrogen, A11120, 1:500). Secondary antibodies: Alexa Fluor-conjugated donkey-anti-mouse and donkey-anti-rabbit (A-21202 and A-21206, Thermo-Fisher, 1:1000), DAPI. All samples were imaged with a Zeiss LSM980 confocal microscope using 25× and 40× oil objective lenses or a Zeiss V16 Axio zoom. We used the same acquisition conditions for all samples per experiment. Fiji (https://imageJ.net/Fiji) was used to assemble all images and measure the mean GFP gut fluorescent. Brightness was adjusted equally across comparable images for clarity.

## Hemolymph extraction and ex vivo brain assay

Hemolymph extraction: we made four holes in the bottom of 0.5 ml Protein LoBind™ Eppendorf tubes and placed each one of them inside

larger 1.5 ml Protein LoBind™ Eppendorf tubes (collection tubes). Next, 50 male flies per sample and genotype were punctured in the abdomen and carefully placed in the small 0.5 ml tube that was kept inside the collection tube. Samples were centrifuged twice at $9000 \times g$ at 4 °C for 5 min. We kept the liquid (hemolymph) in the collection tube and added fresh HL3 buffer (1.5 mM $Ca^{2+}$, 20 mM $MgCl_2$, 5 mM KCl, 70 mM NaCl,10 mM $NaHCO_3$, 5 mM HEPES, 115 mM Sucrose, 5 mM Trehalose) until the total volume reached ~100 µl per sample. Each hemolymph sample was stored at −80 °C and thawed only for use.

Ex vivo brain assay: brains were dissected in HL3 buffer and placed in eight-well clear-bottom cell culture chamber slides with 150 µl HL3. Brains were stabilized with cut paper clips on top of a Nylon mesh (Warner instruments, 64-0198). We used a Zeiss LSM980 microscope with 40× oil objective lens for live imaging. NPF-P1 neurons were located based on the Cherry reporter that NPF neurons expressed together with GCAMP7b. Each frame (~5 s/frame) was the maximum projection of 5 z-stacks (2 µm/z) that were acquired with 491 nm excitation for GFP. After >10 rounds of imaging 100 µl of thawed hemolymph sample was added per brain (Supplementary Movies 1–6). Identical acquisition conditions were used per experiment. Fiji was used for the calculation of peak fluorescent (max ($\Delta F/F_0$)). $\Delta F/F_0 = F_{fr} - F_0/F_0$. $F_{fr}$ is the fluorescence per frame, and $F_0$ (baseline fluorescence) is the average fluorescence intensity of the last 9 frames prior to hemolymph addition.

## Metabolic assays
Four female or 8 male flies per biological replicate, genotype and time-point were collected and stored at −80 °C. Biological replicates were homogenized with 1 mm beads in 300 µl cold 0.1% Triton-X-100 (Sigma X100) using a bullet blender homogenizer (Next Advance). Homogenates were spined down, heat treated for 10 min at 75 °C and 200 µl of lysate per sample was isolated for measurements.

For protein measurements, 5 µl per biological replicate were mixed with 200 µl of BCA Protein Assay Kit (Pierce BCA Protein Assay Kit, Thermo Fisher Scientific, 23225) in 96-well Microplates UV-Star (Greiner Bio-One−655801) and incubated for 30 min at 37 °C with gentle shaking.

The triglyceride content per biological replicate was calculated as the measurement of total glycerol minus the measurement of free glycerol. In detail, for total glycerol, 20 µl per biological replicate were mixed with 150 µl of warmed Infinity triglyceride reagent (Thermo Fisher Scientific, TR22421) in 96-well Microplates UV-star (Greiner Bio-One−655801) and incubated for 10 min at 37 °C with gentle shaking. For free glycerol, 20 µl per biological replicate were mixed with 150 µl of warmed free glycerol reagent (Sigma-Aldrich, F6428) and incubated for 10 min at 37 °C with gentle shaking. The triglyceride content (total glycerol-free glycerol) per biological replicate and genotype was normalized to the control genotype to calculate the fold change.

All quantifications were done with the SpectraMax Paradigm Multi-mode microplate reader (Molecular Devices) using the developer's software. Values were determined from standard curves of serial dilutions of BSA (Pierce BCA Protein Assay Kit) and of glycerol standard solution (Sigma-Aldrich, G7793). Protein measurements were read at $\lambda = 562$ nm. Total glycerol and free glycerol were read at $\lambda = 520$ nm.

## Dry weight quantification
Ten male or female flies per biological replicate, genotype and time-point were anesthetized, placed in 1.5 ml Eppendorf tubes and then heated at 75 °C overnight with closed tube caps. The next day, caps were opened, and biological replicates were left at 75 °C for 2 h (or until completely dry). Dry weight per biological replicate was measured using an analytical balance (Mettler Toledo, AB104-S) and lab weighing paper.

## Lifespan assay
Newly enclosed flies were collected weekly (Monday–Tuesday) and kept at 18 °C (the same temperature that were grown). Lifespan experiments started Friday and were conducted at 29 °C in 12:12 (light-dark) cycle conditions. Male flies were transferred to new standard lab food vials during light hours every 2–3 days (every Monday, Wednesday, and Friday), and death counts were recorded. The lifespan of 9 biological replicates per genotype was recorded. Each biological replicate consisted of 9–16 male flies per genotype. The probability of 75%, 50% (median survival), 25% and 0% survival (Supplementary Fig. 5c) was scored as the day the survival curve passed each survival point, respectively. The probability of death on Fig. 5d shows the percentage of death for each of the 9 biological replicates on day 21 and for each of the 9 biological replicates on day 24 per genotype.

## qPCR transcript measurements
We used the Ambion PureLink RNA Mini Kit (ThermoFisher 12183018 A) to extract RNA per biological replicate, including PureLink DNase (ThermoFisher, 12185010). 500 ng of RNA was amplified and converted to cDNA using the iScript cDNA Synthesis kit (Bio-Rad, 1708890). cDNAs were analyzed using the iQ SYBR Green kit (Bio-Rad, 1708880) and Bio-Rad CFX Manager software. Gene expression was analyzed against a standard curve per gene (absolute quantification). RNA was extracted from 10 to 15 adult male or female flies (whole body); or 15 adult guts; or 30 adult heads per biological replicate. *Rpl32* or *α-tubulin* was used as internal control. Each RT-qPCR was performed with at least three biological replicates. Primers were acquired from IDT, and the exact oligo sequences can be found in Supplementary Data 1.

## Statistical tests and reproducibility
We used Prism 10 (https://www.graphpad.com/scientific-software/prism/) to perform all statistical analysis and to generate all graphs. We conducted Shapiro−Wilk or D' Agostino and Pearson normality tests. Normally distributed data were analysed with unpaired two-tailed t-test or for multiple comparisons with one-way Anova or two-way Anova tests (Sidak's or Tukey's or Dunnett's T3 or uncorrected Fisher's LSD test). Non-normally distributed data were analysed with two-tailed Mann−Whitney test or for multiple comparisons using Kruskal−Wallis with Dunn's test. To compare survival curves for lifespan we used the log-rank (Mantel−Cox) test. All statistical tests are indicated in the figure legends, and exact p-values are shown in figures. Eye-scoring and dry weight assays were done blind. For other experiments, investigators were not blinded to allocation during experiments and outcome assessment. No statistical method was used to predetermine sample size. The experiments were not randomized. No data were excluded from analysis. Confocal images are representative of at least two independent experiments with similar results. All data are generated from two or more independent experiments with at least three biological replicates per genotype and condition. Charts show mean values ± SD for normally distributed data and mean values ± SEM for non-normally distributed data. Boxplots show minimum, 25th percentile, median, 75th percentile, and maximum values.

## Reporting summary
Further information on research design is available in the Nature Portfolio Reporting Summary linked to this article.

# Data availability
All data are available in the main text, Figures, Supplementary Information, Supplementary Data and Supplementary Movies. Source data are provided with this paper. All raw data generated in this study are provided in the Source data file. Correspondence and request for

materials should be addressed to Afroditi.Petsakou@einsteinmed.edu or perrimon@genetics.med.harvard.edu. Source data are provided with this paper.

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

## Acknowledgements

We thank Stephanie Mohr and Tobias Janowitz for comments on the manuscript. Confocal imaging was conducted at the MicRoN Facility at Harvard Medical School. We thank Hugo Bellen, Gerry Rubin, DRSC/TRiP, and the Bloomington Stock Center for fly lines, and Rich Binari for help. Figure 5e was drawn with Adobe Illustrator, and the image on Supplementary Fig. 2c was created with BioRender.com. This work was supported in part by the Cancer Grand Challenges partnership funded by Cancer Research UK (CGCATF-2021/100022) and the National Cancer Institute (OT2 CA278685-01). A.P. was supported by Good Ventures through the Life Science Research Foundation. Mathew Li was supported by the Harvard Stem Cell Internship program. N.P. is an investigator of the Howard Hughes Medical Institute. This article is subject to HHMI's Open Access to Publications policy. HHMI lab heads have previously granted a nonexclusive CC BY 4.0 license to the public and a sublicensable license to HHMI in their research articles. Pursuant to those licenses, the author-accepted manuscript of this article can be made freely available under a CC BY 4.0 license immediately upon publication.

## Author contributions

A.P., N.P. conceived the study. A.P. designed the study. A.P., E.F., Y.C., M.L., and A.Z. performed the experiments. A.P. analyzed the data and wrote the manuscript. A.P., E.F., and N.P. edited the manuscript.

## Competing interests

The authors declare no competing interests.

## Additional information

**Supplementary information** The online version contains Supplementary material available at https://doi.org/10.1038/s41467-026-70074-2.

