## [Transparent Peer Review File · Nature Communications]

Tumor-induced orexigenic imbalance lowers protein appetite and drives early organ wasting symptoms

Corresponding Author: Professor Norbert Perrimon

Version 0:

Reviewer comments:

Reviewer #1

(Remarks to the Author)

The work by Petsakou et al. uses a model of inducible neoplasia ("cancer") in *Drosophila* flies to dissect the early molecular steps leading to changes in feeding behavior, ultimately causing cancer-induced anorexia and organ wasting. The experimental model is well-established and has been previously characterized, primarily through studies spearheaded by the same lab. The presented data reinforce some of the group's previous findings (e.g., the synergy of Upd3 and Impl2 in organ-wasting induction) while substantially expanding earlier observations on the feeding behavior of tumor-bearing flies and providing novel insights into the interplay between cancer-secreted molecules (Upd3 and Impl2) and the brain-secreted neuropeptide NPF.

This work is novel because it details the mechanisms of tumor-induced cachexia and opens new research avenues for investigating similar interactions in clinical settings. Given that the key molecules studied have mammalian homologs, the results described here are of significant interest to broad research and medical communities. Additionally, the reported measurable changes in food preference in flies further bolster the use of the *Drosophila* model in ingestive behavior research. The manuscript is well-written and presents well-documented results.

Major comment:

Perhaps due to the restraints of the short communication format, the manuscript notably lacks a full-fledged discussion that would place the authors' findings in the larger context and, perhaps, speculatively outline a potential mechanism by which Upd3 and Impl2 regulate NPF expression in neurons. While the authors provide some connections selectively, the work would benefit from a more holistic discussion.

Minor comments:

Content:

1. Statistical analysis: please justify why SEM was used instead of SD. SEM might not adequately represent sample distribution. More on the topic, see PMID: 23125963.
2. Additional explanations are required to clarify how TrpA1 expression could "increase NPF signaling".
3. More details on qPCR data analysis in the Methods are required. How was GOI expression normalized? By absolute quantification or double-delta Ct?
4. Fig. 3a: Changes in the expression of several genes (eg. Akh, dilp5, Burs) are not marked as significant, although data point distribution suggests otherwise. Please provide your explanations.
5. Based on figs. 3a and ext. 2a, the authors conclude that putative regulators of hunger/satiety respond differently in males and females. However, no fully comparable datasets support this claim: the female data in Fig. Ext. 2a include a much smaller dataset that does not fully match the male data. Please synchronize the male and female datasets for proper justification.
6. In the final part of the Results, the sentence starting with "Taken together, our data provide evidence that link..." seems to be an overstatement, as the manuscript's data cannot prove that mammalian orthologs of the fly molecules function in the same way. Please consider softening the tone of this statement.
7. Provide more information on how food consumption (fig. ext. 1f) and triglyceride content (fig. ext. 1g) were normalized. Specifically, how was it possible to obtain negative values?

Formatting:

8. Please standardize driver depiction in the Methods (with or without a dash); the canonical form uses a dash b/w enhancer and gene. Most depictions are correct, but some are inconsistent.
9. Figs. 1f, 4b, ext. 1f, ext. 3b,f: What does the dotted line signify? Include its description in figure legends.
10. Fig 2a-b: please mention in the figure legend which method was used for expression quantification.
11. Fig. 3a: Re-center the graph title
12. Consider revising the sentence: "Altogether these prompted us to explore.."
13. Please revise "heat-activated thermosensitive cation channel TrpA1" for possible redundancy
14. There's a typo in the figure reference within the sentence: "NPF activation solely from the gut prevented the rescue of the anorexic behavior in Ykiact male flies (Fig. 4g), suggesting that neuronal NPF signaling is required for the Yki-induced anorexic phenotypes." The correct reference should be Fig. 3g.
15. Please revise "...excitation response, supporting that hemolymph factors..." for clarity.
16. There's a typo in "(Ampa, VWR 45000-614)"
17. Correct "Protein measurements were read at 562λ" and "...520λ" in the next sentence to "Protein measurements were read at λ=562 nm." and "λ=520 nm", respectively.
18. Correct "on" to "in" in: "...from d2 to d6 on control.."
19. Correct "in" to "at" in: "flies are reared until adulthood in low temperatures".

Reviewer #2

(Remarks to the Author)

The study investigates how colorectal tumors disrupt feeding behavior and contribute to early symptoms of cachexia using *Drosophila* as a model system. By employing a well-characterized Yki-induced gut tumor model, the authors examine how tumor-derived factors drive changes in metabolism and appetite before systemic tissue wasting occurs. One of the key findings is that tumor-bearing flies exhibit reduced food intake, even before significant weight loss and organ wasting become apparent. The study identifies Upd3, the *Drosophila* IL-6 homolog, and ImpL2, an IGF1-like factor, as key tumor-secreted molecules that contribute to metabolic dysregulation and feeding suppression. Furthermore, the authors uncover a critical role for neuropeptide F (NPF) in regulating these changes. Their model indicates that tumor growth is associated with reduced NPF expression in the brain, and genetic activation of NPF neurons restores food intake and prevents weight loss, highlighting a tumor-neuroendocrine axis driving cachexia. A particularly intriguing aspect of this study is the selective suppression of protein-rich food intake in tumor-bearing flies. This mirrors findings in cancer patients, where alterations in macronutrient preference are observed during early stages of cachexia. The results suggest that tumor-driven inflammatory and metabolic signaling specifically affects protein appetite, which may have significant implications for understanding the early mechanisms of cancer-related malnutrition.

The manuscript presents interesting observations and provides new insights into the mechanistic pathways through which tumors can induce anorexic effects. Overall, I support the publication of this study, but certain aspects need to be strengthened to fully support the proposed model, in which gut tumors induce upd3 and ImpL2, which in turn inhibit brain NPF to suppress appetite for yeast.

The authors induce gut tumors and observe decreased feeding. They then show that whole-body upd3 and ImpL2 expression is upregulated under these conditions. To test whether upd3 and ImpL2 suppress feeding, they overexpress upd3 and ImpL2 in muscle. However, it is unclear why they chose to overexpress upd3 and ImpL2 in muscle after inducing tumors in the gut, and they do not show whether upd3 and ImpL2 expression actually increases in muscle under tumor conditions. Clarifying this choice and providing supporting evidence would strengthen the study. It is also critical to demonstrate that upd3 and ImpL2 mediate the effects of gut tumors on feeding. This could be addressed, for example, by testing whether gut tumors have no effect on feeding in a upd3/ImpL2 double mutant. If the authors aim to support the idea that muscle-derived upd3 and ImpL2 mediate these effects, they could use the LexA system to simultaneously activate Yki in the gut while knocking down upd3 and ImpL2 in muscle. These epistatic experiments are essential to substantiate the proposed model.

Another part of the model that requires additional support is the claim that muscle-derived upd3 and ImpL2 reduce feeding through suppression of brain NPF. This could be tested using similar epistatic approaches, such as overexpressing upd3 and ImpL2 in muscle while simultaneously manipulating NPF+ cells in the brain using the LexA system. Additionally, the authors use TrpA1 to activate NPF+ cells, but these cells co-express other neuropeptides. To demonstrate that the effects are specifically mediated by NPF, they should simultaneously knock down NPF while activating NPF+ cells with TrpA1. If the effects are truly mediated by NPF, this manipulation should prevent the rescue of feeding suppression. These experiments would directly test whether their proposed pathway is functionally relevant and significantly strengthen the conclusions.

Thus, the data in Figures 3 and 4 support the idea that gut tumors regulate feeding via effects on brain NPF, but additional epistatic experiments are needed to confirm the Upd3/ImpL2 steps in between. There are also some concerns regarding the epistatic experiments in Figure 3. In Figure 3e, activation of NPF+ cells appears epistatic to gut tumors, but in Figure 3d, activation of NPF+ cells promotes feeding in a manner that seems to be additive rather than strictly downstream of gut tumors. A two-way ANOVA could help resolve these issues by identifying interactions between tumor induction and NPF activation, which would clarify whether the effects are truly epistatic.

Additionally, I believe that eye-scoring is not as accurate as measuring intake spectrophotometrically. Is there a specific reason why eye-scoring was chosen instead?

Finally, to further support the conclusions about daily rhythms of feeding behavior, it would be valuable to perform dye assays with either sugar or yeast separately.

One key mechanistic question is whether upd3 is activating JAK-STAT signaling in NPF+ neurons. Have the authors examined the role of the upd3 receptor dome in these neurons? Knockdown of dome in NPF+ cells would provide strong support for this pathway. Similarly, how is ImpL2 acting in this context – does it suppress feeding via an effect on insulin

signaling in NPF+ neurons? Addressing these questions would provide a more complete picture of the molecular mechanisms underlying tumor-induced anorexia.

Reviewer #3

(Remarks to the Author)

A salient feature of cancer cachexia in patient, mouse and fly models is organ atrophy and weight loss. Although experiments from rodent models early on showed that forced feeding does not fully reverse weight loss and organ atrophy, evidence has amassed that substantiates that anorexia is a rate-limiting component of cachexia progression with atrophy and weight loss. The molecular mechanisms mediating cachexia-anorexia are still largely unclear.

In flies, a seminal paper addressing this question in a fly model <https://doi.org/10.1038/s41556-020-00628-z> showed that a tumor model based on Yki-act overexpression in the eye of adult flies induces anorexia through tumor Dilp8 insulin-like peptide secretion that is sensed by Lgr3 receptors in select (NPF) neurons in the brain. Lgr3 and Dilp8 suppress feeding and reduce orexigenic sNPF and NPF production and increases anorexigenic NUCB1 production. Strikingly, the analogous signaling network and peptides are conserved to experimental mouse cancer cachexia models and pancreatic human tumor cells that express the dilp8 orthologue/INSL3 and affect orthologues of NPY/NPF and NUCB2/NUCB1 feeding regulators. The causal relationship of NPF/sNPF was only explored in mice.

In the submitted manuscript by Petsako A, et al, utilizes a "colon cancer" Yki-act model driven by the escargot-GAL4 driver of gut stem cells and lineage and investigates effects on feeding and weight loss. Previous findings from the same lab reported no/little effect on feeding, but more sensitive methods now reveal reduced food intake that precedes a reduction in dry body weight. In this model, NPF is reduced, whereas NUCB1 and sNPF were not measurably different in expression at the whole body level, so the authors conclude that something else may be at play, distinct from eye Yki-act model. The authors chose to focus on NPF neuron regulation and how it mechanistically relates to anorexia & weight loss as this question was not pursued in detail in previous studies.

In a series of impressive experiments combining, sophisticated genetics, gene expression analysis, neuronal activity manipulation, in vitro culture and food preference functional analysis, the manuscript makes a case for a tumor-induced change in food preference that may underlie anorexia and weight loss.

After establishing that Yki-act gut tumors lead to weight loss, the authors ascribe this. Function to the combined expression of the Insulin growth factor binding protein, IMPL2 and Upd3/IL-6, earlier shown to be expressed in Yki-act tumors. This is based on sufficiency experiments by combined ectopic expression of these factors in muscle that reproduce reduced feeding and weight loss effects. Knockdown of these genes in Yki-act tumors is not shown to address if these factors are required in the tumor setting and whether they derive only from tumor. Whether Dilp8/Lgr signaling is involved or acts entirely separate is also not addressed. If raising NPF levels sufficient/insufficient to rescue the defects is also not experimentally addressed.

Instead, the authors show that NPF neuronal activity is a likely mediator. Combined sophisticated imaging and manipulation of NPF neurons' activity show that NPF activity is lower when exposed to hemolymph from tumor-carrying animals and that stimulating NPF signaling in vivo rectifies feeding and food choice defects. Whether IMPL-2 and Upd3 directly affect NPF neurons or whether this is mediated through other secondary effectors or metabolites is not addressed.

By manipulating NPF activity, it is shown that the tumor-induced reduced preference for protein rich food can be rectified through increasing NPF neuronal signaling. The downstream mechanisms for this behavioral effect are not addressed.

Experiments are well executed with relevant controls and statistics according to the standard of the field.

Altogether, the manuscript invokes reduced NPF neuronal activity in regulating feeding and food choice in response to tumor produced factors. They make a convincing argument that NPF activity, possibly in addition to reduced NPF expression, controls food choice and ingestion, but their relative contribution is not addressed.

Open questions remains and needs to be addressed:

1. Is Dilp8-LGR3 signaling involved in NPF neuron signaling and regulating NPF levels as suggested by the eye Yki-act tumor model? Experiments to include or exclude Dilp/Lgr3 should be shown. I.e. does Lgr3 knockdown in NPY neurons affect neuronal activity/food choice caused by Yki gut tumors? Is Dilp8 increased in Yki-act gut tumor models?
2. How does reduced NPF levels relate to NPF neuronal activity and food preference? Are these parallel events, and does NPF levels matter? A rescue of NPF levels in NPF neurons could be experimentally addressed in Yki-act animals. Can the authors disentangle NPF activity from NPF production?
3. Upd3-IMPL2 both are needed to target NPF neuronal activity. Whether this is due to direct control or if indirect mechanisms are at play is left unaddressed. Previous work has shown that both Upd1 and S6K status (downstream in Insulin-TORC1 signaling) can regulate NPF neurons and feeding. Addressing if the Upd receptor (Dome) and InR-PI3K-TORC1-S6K signaling affects NPF neuronal activity could be experimentally addressed to see if any of the mediators act directly on the NPF neuron through IMPL2 and Upd3.
4. Why is Upd3/IMPL2 double knockdown not shown for tumor tissue? This is important and would be necessary to argue

that the tumor secretes these factors to regulate NPF neurons and in turn regulate feeding. It is possible that ImpL2 and Upd3 expression in response to Yki-Act tumors is broader than the tumor itself, furloughing direct tumor knockdown experiments. If this is the case, this should be shown by using existing reporters (preferred), IF analysis or qPCR. Or is the Upd3/IMPL2 effect thought to act indirectly through another mechanism?

Specific comments:

Fig1 It seems like the *esgTs>GFP* have higher expression in the males in general (brain and gut). Is it true that there is sex differential *esg* expression, or is it only a trend in these pictures? Could these explain why the feeding phenotype is more dramatic in the males? . No quantifications are shown for tumour size when gut tumours are shown. Can the authors provide tumor volumes for a given gut region to make a firm statement on the effects on tumor growth?

Fig2 a-b. Why is whole body expression assessed for ImpL2 and *upd3* rather than from tumor/gut? From previous work, the expectation is that these secreted proteins originate from tumor cells. Do the authors expect or observe otherwise? Reporters for *Upd3* and IMPL2 exist. Where are they expressed in control and tumor conditions? Are their expression controlled by starvation? 2d-e. While it is interesting that the combined effect of ImpL2 and *Upd3* expression from muscle, lead to reduced feeding, it is unclear why this approach is chosen. Why not express using *Mex-GAL4* (gut) for instance, which approximates the tumor model in location and possibly cell type(s). Are *Upd3* and ImpL2 expressed in muscle upon 1. gut tumor presence or 2. starvation? To make a firm case that tumor-produced *Upd3* and IMPL2 mediates anorexia, the authors should provide experiments where these factors are knocked down in tumors. The reduced growth of tumors upon *Upd3* knockdown can be counteracted by Hop-Act in tumors for instance if reduced tumor growth is a caveat. Epistasis experiments in NPF neurons to show if *Upd3* and IMPL2 acts directly on NPF neurons could be addressed by manipulating *Dome* and *InR-S6K* signaling. Why are these experiments not pursued

Fig3

In the previous *eye-Yki-act* model studies (reference 37), they used females to study anorexia. Did the authors address if this is the reason for differences in *sNPF* and *nucb1* between the studies?

g – Why are there so many dots on the graph that are on 0? How long were these bouts? Why are they never seen or seen very little on the control column in all flypad graphs? Should there be a filter to what is considered a bout? Why is there a difference with the third group? If there is no tumour, and there is no contribution from the gut NPF, why do these flies eat less?

h- Why do the authors study NPF neuronal activation at D6 and not at D4, as the expression drop was found at D4 in fig. 3b

Fig4

a – It would be favorable if the authors could provide tumour volumetrics.

Optional/suggestion:

- Survival curves for rescues of anorexia. It would be interesting if there is a link between NPF neuronal activity rescue and whether it leads to increased animal survival through reversal of anorexia. This may, however, be beyond the scope of this work.

Version 1:

Reviewer comments:

Reviewer #1

(Remarks to the Author)

The updated manuscript has undergone substantial revision, strengthening the previous findings, providing additional mechanistic insight into the synergy between ImpL2 and *upd3*, and adding new survival data. The authors have expanded the female dataset to justify their focus on males. These changes significantly improve the manuscript and address the reviewer's earlier concerns. However, several major issues remain to be resolved.

Main Points

1. This point should have been raised in the first review round but was inadvertently omitted from my summary. There is a concern that the authors overly "humanize" the fruit fly model. While the reviewer supports efforts to emphasize the relevance of *Drosophila*-based research, in some cases the interpretation extends too far. Specifically, referring to experimentally induced midgut neoplasia as "colorectal tumors" requires reconsideration. Although the midgut is part of the fly gastrointestinal tract, it is not functionally or morphologically equivalent to the mammalian colon or rectum. The reviewer understands the authors' intent to engage the medical community; however, equating the midgut with the colon is misleading. The authors are encouraged to provide a clear and defensible justification for using the term "colorectal" in the context of their study.

Similarly, when discussing the role of blood-brain barrier weakening in enabling neuronal expression of insulin-like hormones during cancer cachexia, the authors should include a disclaimer acknowledging that this mechanism is specific to insects and other invertebrates and cannot be directly extrapolated to vertebrates, in which insulin and insulin-like peptides are produced by non-neuronal tissues. Such transparency will be valuable to readers navigating the complex and sometimes inconsistent literature on cancer cachexia.

2. The conclusions drawn from Fig. 5c–d require careful reconsideration. The mortality differences presented at 50% and 0% survival (Fig. 5c) are not as clear as implied. Additional checkpoints at 25% and 75% survival would likely show that the differences between the *elavGal80;eagTS>yki; ±TrpA1* groups are not statistically significant.

In Fig. 5d, the comparison between the two rightmost groups (i.e., *elavGal80;eagTS>yki; NPFLexA>+* and *elavGal80;eagTS>yki; NPFLexA>TrpA1*) cannot reasonably yield a p-value < 0.001 as indicated by the triple asterisks. Although the reviewer does not have access to the raw data, simulating datapoints from the plotted values produces a Student's t-test p-value > 0.05. Given these limitations, presenting NPF alterations as the primary explanation for improved survival appears premature.

3. Interpretation of male vs. female datasets. Now that comparable datasets are available, the female data do not appear substantially different from the male data. Both NPF and *ccha2* expression in females show the same directional trends as in males, even if not highlighted as significant. As the authors note in their rebuttal, adjusting statistical methods could yield significance for factors that visually differ between control and *yki+* conditions. In fact, aside from the increase in *upd2*, few features distinguish the female dataset. In this context, the authors' insistence on treating males separately seems insufficiently supported. A more straightforward solution may be to combine the male and female datasets into a unified analysis, which would simplify interpretation and eliminate the added complexity of sex-specific handling.

Minor Points

4. Add male/female designations to all supplementary figure panels (Figs 1, 2, supple 1, and supple 2), as done in Supplementary Figure 2a, to prevent confusion for readers encountering similar datasets across different figures.

5. The Methods section does not specify how the flies were aged for the experiment shown in Fig. 5d. Please include this information.

6. Line 382: the word "be" is missing and should be added.

Reviewer #2

(Remarks to the Author)

The authors have adequately addressed my previous concerns. The added epistatic and mechanistic experiments, together with clearer methodological justification, substantially strengthen the manuscript.

Reviewer #3

(Remarks to the Author)

The revised manuscript by Petsakou A et al. has been considerably reworked and improved.

The authors addressed all my concerns to my satisfaction.

Version 2:

Reviewer comments:

Reviewer #1

(Remarks to the Author)

Thank you to the authors for addressing my points and providing thoughtful responses. While some data — particularly those related to survivorship — remain open to interpretation, the reviewer concurs with the authors' position and do not wish to press these points further.

Response to referees:

We thank all 3 referees for their comments and support. Their constructive comments have guided us in adding new experiments to strengthen our finding that the synergistic function of two conserved tumor-secreted factors cause an early orexigenic imbalance in the brain that drives organ wasting symptoms. In addition, we have increased the length of the revised manuscript to a full article to better clarify our reasoning for the experiments and to provide an in-depth discussion as to how our findings advance our current understanding of early-stage cancer cachexia.

The main concern of the referees was that our previous manuscript didn't provide a mechanism as to how the combined function of tumor-derived upd3/IL-6 and Impl2/IGFBP leads to anorexia prior to organ wasting in colorectal Yki-tumor male flies. We have now added new epistatic and knock down experiments that address this. Overall, our new experiments support that excess of Impl2 reduces neuronal NPF signaling and the synergistic role of upd3 is to strengthen the inhibitory function of Impl2 leading to anorexia. Based on our findings we propose that upd3 helps Impl2 by disrupting the BBB and allowing aberrant access to the brain.

Another concern included differences in feeding-related signals of male and female colorectal Yki-tumor flies. We have now re-organized our manuscript to focus more on the mechanism of male flies, while also providing a more complete dataset on the changes of feeding-related signals in female colorectal Yki-tumor flies. In addition, we were asked about the overall impact of early orexigenic imbalance to organ wasting. We have now added a completely new set of experiments supporting that early rescue of NPF-linked orexigenic imbalance reduces the high death risk of colorectal Yki-tumor flies.

Below is our detailed answer to each referee. Black wording is our answer, and blue wording is the question of each referee. In addition, in the revised manuscript we have used red wording to highlight sentences that directly address reviewer comments.

REVIEWER COMMENTS

Reviewer #1 (Remarks to the Author):

The work by Petsakou et al. uses a model of inducible neoplasia ("cancer") in Drosophila flies to dissect the early molecular steps leading to changes in feeding behavior, ultimately causing cancer-induced anorexia and organ wasting. The experimental model is well-established and has been previously characterized, primarily through studies spearheaded by the same lab. The presented data reinforce some of the group's previous findings (e.g., the synergy of Upd3 and Impl2 in organ-wasting induction) while substantially expanding earlier observations on the feeding behavior of tumor-bearing flies and providing novel insights into the interplay between cancer-secreted molecules (Upd3 and Impl2) and the brain-secreted neuropeptide NPF.

This work is novel because it details the mechanisms of tumor-induced cachexia and opens new research avenues for investigating similar interactions in clinical settings. Given that the key molecules studied have mammalian homologs, the results described here are of significant interest to broad research and medical communities. Additionally, the reported measurable changes in food preference in flies further bolster the use of the Drosophila model in ingestive behavior research. The manuscript is well-written and presents well-documented results.

We thank the reviewers for the support! We have now revised the manuscript based on the reviewer comments. We have answered each comment and made the appropriate changes in the manuscript as explained below in detail.

Major comment:

Perhaps due to the restraints of the short communication format, the manuscript notably lacks a full-fledged discussion that would place the authors' findings in the larger context and, perhaps, speculatively outline a potential mechanism by which Upd3 and Impl2 regulate NPF expression in neurons. While the authors provide some connections selectively, the work would benefit from a more holistic discussion.

We thank the reviewer for this comment. We have now changed the format of the manuscript to an article and included a more holistic discussion that places our findings in the larger context.

We also added new epistatic experiments that strengthen the role of upd3 and Impl2 in tumor-induced anorexia and based on these experiments proposed a potential mechanism. Specifically, in Yki-tumor flies we knocked down *Impl2* in the gut (Fig. 3b-c) and upd3 signaling in the blood brain barrier (BBB, Ext. Data. Sup. Fig. 3e) both of which rescued the feeding behavior. We also tested the *NPF* levels in these flies (Fig. 3d, Ext. Data Sup. Fig. 3f) and next tested whether elevated amounts of Impl2 or upd3 alone impact the activity of NPF neurons using an *ex vivo* brain assay (Fig. 3h, Ext. Data. Sup. Fig. 3g-j).

Based on these experiments we propose in the manuscript: *“Altogether, our data suggest that excessive increase of Impl2 drives the reduction of neuronal NPF signaling and that upd3 augments the inhibitory impact of Impl2, leading to anorexia. We propose that upd3 disrupts the BBB and increases passage of Impl2 to the brain”.*

Minor comments:

Content:

1. Statistical analysis: please justify why SEM was used instead of SD. SEM might not adequately represent sample distribution. More on the topic, see PMID: 23125963.

We thank the reviewer for this comment. As per reviewer recommendation we edited some of our graphs. Specifically, when data were normally distributed (using normality tests, see Methods) we changed the errors in the graphs from SEM to SD. However, when our data were not normally distributed (e.g. graphs showing total duration of activity bout) we kept SEM, since SD is not a good measure of variability as reported in the following study: “Exceptionally, the SD as an index of variability may be a deceptive one in many experimental situations where biological variable differs grossly from a normal distribution (PMID: 23125963)”.

We have revised our Method sections (Statistical tests and reproducibility) by writing: *“Charts show mean values \pm SD for normally distributed data and mean values \pm SEM for non-normally distributed data.”*

2. Additional explanations are required to clarify how TrpA1 expression could “increase NPF signaling”.

We thank the reviewer for this comment. We have now added the following clarification regarding the function of TrpA1 and have added references that used TrpA1 to elevate NPF signaling in NPF-expressing neurons and enteroendocrine cells: We revised our manuscript and wrote: *“Specifically, at high temperature (~29°C) TrpA1 allows the influx of cations inside the cell which increases activation and release of NPF from NPF-expressing neurons and enteroendocrine gut cells (PMC9684077, PMC8355161, PMC10743384, PMC5536846, PMC5617300).*

3. More details on qPCR data analysis in the Methods are required. How was GOI expression normalized? By absolute quantification or double-delta Ct?

We thank the reviewer for this comment. We have now added the following clarification in the Method section (qPCR transcript measurements) regarding the way qPCR data were analyzed: “Gene expression was analyzed against a standard curve per gene (absolute quantification).”

4. Fig. 3a: Changes in the expression of several genes (e.g. Akh, dilp5, Burs) are not marked as significant, although data point distribution suggests otherwise. Please provide your explanations.

We thank the reviewer for this comment. To identify the most significantly differentially expressed feeding-peptides among the multiple related genes in Yki^{act} flies we used a statistical test with the highest power, the two-way Anova using Sidak’s multiple comparison test. This test corrects for multiple comparisons and for false positives (PMID:30157585). Even when conducting another commonly used test for multiple comparisons, like the two-way Anova Bonferroni’s multiple comparison test, the significantly differentially expressed genes remained the same (*NPF* and *ccha2* in male flies (Fig. 2a) and *upd2* in female control vs Yki^{act} flies on d5 (Ext. Data. Sup. Fig. 2a)). Here are the detailed non-significant p-values for *Akh*, *dilp5* and *Burs* in males based on two-way Anova Sidak’s and Bonferroni’s multiple comparison test respectively: *Akh* (p-value:0.5299 & 0.7430), *dilp5* (p-value:0.7039 & 0.999), *Burs* (p-value: 0.1378 & 0.1478).

As the reviewer correctly observed, several genes showed changes in expressions albeit not significant based on two-way Anova Sidak’s and Bonferroni’s multiple comparison test. To this end, if we change our statistical test to a less stringent test like the two-way Anova uncorrected Fisher’s LSD multiple comparison test, that doesn’t consider the multiple comparisons and therefore has higher chances of showing false positives, then we can detect p-values <0.05 in more genes. Given that our goal was to primarily focus on genes exhibiting the most significant changes, we are showing the p-values using the Sidak’s multiple comparison test.

However, we changed our wording to acknowledge that several genes were differentially expressed albeit not as strongly. Specifically, we wrote : “During the onset of anorexia, we observed several feeding-related signals to have variable expression in Yki^{act} males, with CCHamide-2 (*ccha2*) and *NPF* showing the strongest statistical decrease (Fig. 2a).”

5. Based on figs. 3a and ext. 2a, the authors conclude that putative regulators of hunger/satiety respond differently in males and females. However, no fully comparable datasets support this claim: the female data in Fig. Ext. 2a include a much smaller dataset that does not fully match the male data. Please synchronize the male and female datasets for proper justification.

We thank the reviewer for this comment. We have now added the same datasets in male and females.

Our conclusions do not differ. We still observe differences between female and male datasets. The strongest one in the female dataset is the main satiety regulator *upd2/leptin*, that is significantly increased in females and not significantly different in males (Fig.2a and Extended data Sup. Fig. 2a). The significant increase of *upd2* in female Yki^{act} flies (p-value: 0.0036 two-way Anova Sidak’s multiple comparison test & 0.0001 two-way Anova uncorrected Fisher’s LSD) and the non-significant increase in male Yki^{act} flies (p-value: 0.9999 two-way Anova Sidak’s multiple comparison test & 0.6690 two-way Anova uncorrected Fisher’s LSD) was consistent among different statistical tests.

6. In the final part of the Results, the sentence starting with “Taken together, our data provide

evidence that link...” seems to be an overstatement, as the manuscript’s data cannot prove that mammalian orthologs of the fly molecules function in the same way. Please consider softening the tone of this statement.

We thank the reviewer for this comment. Taking into consideration the reviewer’s suggestion we have lowered the tone of the entire manuscript, provided references on how our findings align with mammalian CC reports and edited the above statement to: “Overall, this work advances our current knowledge of events occurring prior to organ wasting by linking tumor-induced i) inflammation, blood brain barrier impairment and reduction of insulin signaling to the advent of anorexia prior to organ wasting; ii) NPF dysfunction to early weight loss and high death risk; iii) decrease of protein-specific appetites to the onset of organ wasting.

7. Provide more information on how food consumption (fig. ext. 1f) and triglyceride content (fig. ext. 1g) were normalized. Specifically, how was it possible to obtain negative values?

We thank the reviewer for this comment.

The value of triglycerides is calculated as (total glycerol - free glycerol) / total protein per genotype per day. The value is the average of 3 samples per genotype per day. To calculate the fold change of triglycerides we normalized to the control sample per day. The reason that values in Yki^{act} samples at d6 are close to zero (Fig. 1k, Ext. Data Sup. Fig. 5a) is because the values of free glycerol were higher in some samples.

We have updated the figure legends and Method section to make clearer how we calculated the triglyceride content.

Figure legend 1k & Ext. Data Sup. Fig. 5a: “....Graph shows triglyceride content (total glycerol – free glycerol levels, see methods) over protein levels. Normalized to control genotype...”

Method section (Metabolic assays): “The triglyceride content per sample was calculated as the measurement of total glycerol minus the measurement of free glycerol as described previously (PMC12021660, PMC4437243). In detail, for total glycerol, 20ml per sample were mixed with 150ml of warmed Infinity Triglyceride Reagent (Thermo Fisher Scientific, TR22421) in 96-well Microplates UV-Star (Greiner Bio-One – 655801) and incubated for 10 min at 37°C with gentle shaking. For free glycerol, 20ml per sample were mixed with 150ml of warmed Free Glycerol Reagent (Sigma-Aldrich, F6428) and incubated for 10 min at 37°C with gentle shaking. The triglyceride content (total glycerol - free glycerol) per sample was normalized to the sample with the control genotype to calculate the fold change.”

For food consumption, we always use a blank sample per genotype that we treat equally to our experimental sample (same rearing conditions, tumor-induction day, age, gender etc.) except that for the blank sample we do not add blue dye in the food of the flies. For each genotype we test, we subtract the value of the blank sample, to account for background differences per genotype. We clarify that in the Methods by writing: “For each different genotype we included a blank sample (no blue dye in the food, 20 flies per blank sample) that was treated identically to account for background differences. Food consumption was calculated by subtracting the value of the blank sample per genotype. To calculate the fold-change the value of food consumption for each genotype (after the blank was subtracted) was normalized to the control genotype.”

Formatting:

8. Please standardize driver depiction in the Methods (with or without a dash); the canonical form uses a dash b/w enhancer and gene. Most depictions are correct, but some are inconsistent.

We thank the reviewer for this comment.

We use dash when a full description is required specifically when: i) first introducing a driver in the main txt; ii) in the method section (Fly stocks); and iii) in the supplementary information when we write the detailed version of each genotype per figure.

We use the version without dash to shorten our writing: in i) figure legends; ii) figures and iii) when writing genotypes in the main txt after we have first introduced them and provided the shortened version.

9. Figs. 1f, 4b, ext. 1f, ext. 3b,f: What does the dotted line signify? Include its description in figure legends.

We thank the reviewer for this comment. We have now included a description of each dotted line in the figure legends.

The dotted line either marks the zero line and negative values, or when showing the dry weight or *NPF* levels helps the reader to easily compare against *Yki^{act}* samples and visualize the 0.75 and 0.6 line respectively.

10. Fig 2a-b: please mention in the figure legend which method was used for expression quantification.

We thank the reviewer for this comment. We have now included a more detailed description in the method section and updated the figure legend in revised Fig. 2a and Extended Data Sup. Fig. 2a as: “To quantify the expression of each gene we generated standard curves and used absolute quantification (see Methods)”.

11. Fig. 3a: Re-center the graph title

We thank the reviewer for this comment. We have now re-centered all graph titles

12. Consider revising the sentence: “Altogether these prompted us to explore..”

We thank the reviewer for this comment. We revised the sentence to: “Since the function of *NPF* in tumor-induced anorexia and organ wasting remains unclear, we explored its role in the anorexic behavior of male *Yki^{act}* flies.”

*13. Please revise “heat-activated thermosensitive cation channel *TrpA1*” for possible redundancy*

We thank the reviewer for this comment. We removed the word “heat-activated”.

*14. There’s a typo in the figure reference within the sentence: “NPF activation solely from the gut prevented the rescue of the anorexic behavior in *Yki^{act}* male flies (Fig. 4g), suggesting that neuronal NPF signaling is required for the *Yki*-induced anorexic phenotypes.” The correct reference should be Fig. 3g.*

We thank the reviewer for this comment. We have now more carefully referenced our figures.

15. Please revise “...excitation response, supporting that hemolymph factors...” for clarity.

We thank the reviewer for this comment. We have now revised the sentence to: “Interestingly, adding control hemolymph to *NPF-P1* neurons caused a post-inhibitory excitation response. This excitation response indicates that secreted factors in control hemolymph allow neuronal activation”

16. There’s a typo in “(Ampa, VWR 45000-614)”

We thank the reviewer for this comment. We have now edited the txt to: “VWR 45000-614”

17. Correct “Protein measurements were read at 562λ” and “...520λ” in the next sentence to “Protein measurements were read at λ=562 nm.” and “λ=520 nm”, respectively.

We thank the reviewer for this comment. We have now corrected the sentence as instructed.

18. Correct “on” to “in” in: “...from d2 to d6 on control..”

We thank the reviewer for this comment. We have now corrected it as instructed.

19. Correct “in” to “at” in: “flies are reared until adulthood in low temperatures”.

We thank the reviewer for this comment. We have now corrected it as instructed.

Reviewer #2 (Remarks to the Author):

The study investigates how colorectal tumors disrupt feeding behavior and contribute to early symptoms of cachexia using Drosophila as a model system. By employing a well-characterized Yki-induced gut tumor model, the authors examine how tumor-derived factors drive changes in metabolism and appetite before systemic tissue wasting occurs. One of the key findings is that tumor-bearing flies exhibit reduced food intake, even before significant weight loss and organ wasting become apparent. The study identifies Upd3, the Drosophila IL-6 homolog, and ImpL2, an IGFBP-like factor, as key tumor-secreted molecules that contribute to metabolic dysregulation and feeding suppression. Furthermore, the authors uncover a critical role for neuropeptide F (NPF) in regulating these changes. Their model indicate that tumor growth is associated with reduced NPF expression in the brain, and genetic activation of NPF neurons restores food intake and prevents weight loss, highlighting a tumor-neuroendocrine axis driving cachexia. A particularly intriguing aspect of this study is the selective suppression of protein-rich food intake in tumor-bearing flies. This mirrors findings in cancer patients, where alterations in macronutrient preference are observed during early stages of cachexia. The results suggest that tumor-driven inflammatory and metabolic signaling specifically affects protein appetite, which may have significant implications for understanding the early mechanisms of cancer-related malnutrition.

The manuscript presents interesting observations and provides new insights into the mechanistic pathways through which tumors can induce anorexic effects. Overall, I support the publication of this study, but certain aspects need to be strengthened to fully support the proposed model, in which gut tumors induce upd3 and ImpL2, which in turn inhibit brain NPF to suppress appetite for yeast.

We thank the reviewer for the support! We have now strengthened our model as per reviewer request and addressed the reviewer comments. Our detailed answers to the reviewer’s questions (underlined) are below.

The authors induce gut tumors and observe decreased feeding. They then show that whole-body upd3 and ImpL2 expression is upregulated under these conditions. To test whether upd3 and ImpL2 suppress feeding, they overexpress upd3 and ImpL2 in muscle. However, it is unclear why they chose to overexpress upd3 and ImpL2 in muscle after inducing tumors in the gut, and they do not show whether upd3 and ImpL2 expression actually increases in muscle under tumor conditions. Clarifying this choice and providing supporting evidence would strengthen the study.

We thank the reviewer for this comment. We have changed the length of the manuscript to a full article and have now better clarified why we chose to overexpress these signals in the muscle.

We used this approach as it was previously used by the Bilder lab (PMC4390765) to specifically assess the impact of a secreted factor (ImpL2) in organ wasting while avoiding the background effect of the tumor growth. In this study, large tissues like the muscle were chosen so that the secreted factor would be released in excess in the fly hemolymph (blood), reminiscent of the high amounts during tumor growth (PMC4390765). The advantage of this approach is that you can particularly assess what is the impact of a specific signal without the indirect effect of additional tumor-induced factors and tumor-induced complications.

Another reason why we chose the muscle as a large tissue is because previous work by the Song lab reported that during colorectal Yki-tumor growth ImpL2 is released from the gut but is also supplementary released from the muscle which amplifies the excessive amount ImpL2 secreted in the hemolymph (PMC8410949).

To clarify all these, we have now revised our manuscript as such: *“Previous work tested the role of ImpL2 in organ wasting independent of tumor growth by ectopically over-expressing ImpL2 in large tissues such as the muscle, and observed that excessive circulation of ImpL2 in tumor-free flies is sufficient to promote organ wasting(PMC4390765). To test for a role of ImpL2 in anorexia independent of colorectal tumor growth, we took a similar approach and conditionally overexpressed ImpL2 from a large tissue. We chose the muscle because it was used previously (PMC4390765) and because in colorectal Yki-tumor flies elevated amounts of circulating ImpL2 originate primarily from the gut but have also been reported to come from the muscle (PMC8410949).”*

Finally, we have strengthened the role of ImpL2 and upd3 as per reviewer request (revised Figure 3) with experiments that we describe in detail below while answering other questions of the reviewer.

It is also critical to demonstrate that upd3 and ImpL2 mediate the effects of gut tumors on feeding. This could be addressed, for example, by testing whether gut tumors have no effect on feeding in a upd3/ImpL2 double mutant. If the authors aim to support the idea that muscle-derived upd3 and ImpL2 mediate these effects, they could use the LexA system to simultaneously activate Yki in the gut while knocking down upd3 and ImpL2 in muscle. These epistatic experiments are essential to substantiate the proposed model.

We thank the reviewer for this comment. We have now added new epistatic experiments as per reviewer request that further strengthen our model and the role of upd3 and ImpL2 signaling in the anorexic behavior of colorectal Yki-tumor flies. Specifically,

- 1) we knocked down *ImpL2* in the gut in Yki^{act} male flies at d5-d6 which rescued both anorexia and *NPF* expression, to levels similar to control (Fig. 3b-d). This suggests that tumor-induced ImpL2 signaling is required for reduction of *NPF* expression and subsequent decrease in feeding behavior.
- 2) we knocked down *upd3* in the gut in Yki^{act} male flies, which rescued feeding behavior (Ext. Data. Sup. Fig. 3b). However, as we explain in the manuscript, this result is difficult to interpret. Previous work has reported that knocking down *upd3* in the gut prevents Yki-tumor growth (PMC8410949). Therefore, the rescue in feeding behavior that we observe when we knock down *upd3* in the gut of Yki^{act} flies, is not a direct effect in feeding-related signals but is caused by the inhibition of colorectal tumor growth itself.
- 3) we combined the LexA and Gal4 system (as per reviewer request) to reduce upd3 signaling in the blood brain barrier (BBB) while the Yki-tumor was growing in the gut of male flies. We found that reduction of upd3 signaling in the BBB during colorectal Yki-tumor growth significantly increased food intake (Ext. Data. Sup. Fig. 3e) and also subtly elevating *NPF* expression (Ext. Data. Sup. Fig. 3f). Previous work supports that in tumor-induced flies elevated upd3 signaling disrupts the BBB leading to aberrant permeability and passage to brain (PMC8511098). Therefore, our findings point to a role for upd3-induced disruption of BBB permeability in tumor-induced anorexia.

Taken together, these epistatic experiments support that upd3 and ImpL2 mediate the effects of gut tumors on feeding. ImpL2 drives anorexia and upd3 helps by disrupting the BBB.

Moreover, to address another of the reviewer point " ... *If the authors aim to support the idea that muscle-derived upd3 and ImpL2 mediate these effects.....*" .

We do not propose that in colorectal Yki-tumor flies upd3 originates from the muscle. We apologize to the reviewer if we didn't make that clearer in the first version of the manuscript that was shorter. As we answered in a previous comment made by the reviewer we overexpressed upd3 and ImpL2 from the muscle as a way to assess the individual role of ImpL2 or upd3 or their combined role in feeding behavior without the indirect effect of additional tumor-induced factors and tumor-induced complications.

Another part of the model that requires additional support is the claim that muscle-derived upd3 and ImpL2 reduce feeding through suppression of brain NPF. This could be tested using similar epistatic approaches, such as overexpressing upd3 and ImpL2 in muscle while simultaneously manipulating NPF+ cells in the brain using the LexA system.

We thank the reviewer for this comment. We have added experiments that strengthen the role of ImpL2 and upd3 in neuronal NPF signaling independent of tumor growth.

Specifically:

1) we assessed whether excessive increase of ImpL2 or upd3 or both in the hemolymph (fly blood) impacts the activation of NPF-P1 neurons using an *ex vivo* brain assay (Fig. 3h, Ext. Data. Sup. Fig. 3g-j). We found that adding hemolymph from flies with excessive increase of ImpL2, inhibited the activity of NPF neurons (Fig. 3h, Ext. Data. Sup. Fig. 3h,3j). However, the activity of NPF neurons was not inhibited when adding hemolymph from flies with excessive increase of upd3 (Fig. 3h, Ext. Data. Sup. Fig. 3g,3j). Finally, adding hemolymph from flies with excessive increase of ImpL2 and upd3, inhibited the activity of NPF neurons (Fig. 3h, Ext. Data. Sup. Fig. 3i-j).

2) we tested the levels of *NPF* expression in the head and whole body of flies with excessive increase of ImpL2 and upd3 and found that *NPF* is significantly decreased to levels similar to *NPF* reduction in Yki^{act} flies (~40% decrease, Ext. Data. Sup. Fig. 3d, Fig. 3g). However, excessive increase of upd3 alone didn't change the expression of *NPF* and excessive increase of ImpL2 alone caused only a moderate 20% decrease of *NPF* expression (Fig 3g).

Together, these results suggest that ImpL2 and not upd3 is responsible for the inhibition of neuronal NPF signaling, and that the inhibitory role of ImpL2 is amplified when upd3 is in excess.

Additionally, the authors use TrpA1 to activate NPF+ cells, but these cells co-express other neuropeptides. To demonstrate that the effects are specifically mediated by NPF, they should simultaneously knock down NPF while activating NPF+ cells with TrpA1. If the effects are truly mediated by NPF, this manipulation should prevent the rescue of feeding suppression. These experiments would directly test whether their proposed pathway is functionally relevant and significantly strengthen the conclusions.

We thank the reviewer for this comment. The reviewer is correct, and we tried to perform this experiment. However, the already packed transgenic load in Yki-tumor flies with TrpA1-induced NPF activation, made impossible to generate male flies with an additional *NPF-RNAi* transgene. Still, there are several published works that activate NPF+ cells with TrpA1 and then use *NPF-RNAi* to verify that TrpA1 was indeed increasing NPF signaling (PMC9684077, PMC8355161, PMC10743384, PMC5536846, PMC5617300). We have now referenced these studies in our revised manuscript. We wrote: "*Specifically, at high temperature (~29°C) TrpA1 allows the influx of cations inside the cell which increases activation and release of NPF from*

NPF-expressing neurons and enteroendocrine gut cells (PMC9684077, PMC8355161, PMC10743384, PMC5536846, PMC5617300)”.

Thus, the data in Figures 3 and 4 support the idea that gut tumors regulate feeding via effects on brain NPF, but additional epistatic experiments are needed to confirm the *Upd3/Impl2* steps in between. There are also some concerns regarding the epistatic experiments in Figure 3. In Figure 3e, activation of NPF+ cells appears epistatic to gut tumors, but in Figure 3d, activation of NPF+ cells promotes feeding in a manner that seems to be additive rather than strictly downstream of gut tumors. A two-way ANOVA could help resolve these issues by identifying interactions between tumor induction and NPF activation, which would clarify whether the effects are truly epistatic.

We thank the reviewer for this comment. As the reviewer requested we performed two-way Anova using Sidak’s multiple comparison’s test for the data in Fig. 2d (previous Figure 3d). Below is the new graph showing the two-way Anova comparisons (same data as revised Fig. 2d but re-arranged for better visualization). Our results indicate that the effect is epistatic and not additive.

Below are the specific genotypes illustrated in the graph:

- i) no tumor + No-TrpA1-NPF activation = *elavGa80; esg^{TS}>+; NPFlexA>+*
- ii) no tumor + Yes-TrpA1-NPF activation = *elavGa80; esg^{TS}>+; NPFlexA>LexAopTrpA1*
- iii) tumor + No-TrpA1- NPF activation = *elavGa80; esg^{TS}>Yki^{act}; NPFlexA>+*
- iv) tumor + Yes-TrpA1- NPF activation = *elavGa80; esg^{TS}>Yki^{act}; NPFlexA> LexAopTrpA1*

Reviewer Fig 1: Graph shows feeding index of the same experiment and data as described in Fig. 2d. Statistics were conducted with two-way Anova using Sidak’s multiple comparison test (GraphPad Prism). Exact p-values are shown in the figure. Chart shows mean values ± SD.

Additionally, I believe that eye-scoring is not as accurate as measuring intake spectrophotometrically. Is there a specific reason why eye-scoring was chosen instead?

We thank the reviewer for this comment. Eye-scoring requires a person to score flies which by definition is permissive to human error. That is why we accompany eye-scoring with spectrophotometry or FlyPAD that provide values in an automated manner. Despite that, our eye-scoring results always aligned with the results from spectrophotometry or FlyPAD which is why we use this method.

In addition, eye-scoring is really helpful in situations that some flexibility is required, e.g. when testing flies with multiple different genotypes and one genotype has a more limited sample size. Spectrophotometry requires exactly 20 flies per sample, approximately 4-5 samples per genotype including always an extra blank sample with additional 20 flies per genotype. Instead, eye-scoring is more flexible in the size of the sample as long as all genotypes tested at each round have the same sample size. We use between ~15-20 flies per sample and test 3-4 samples per genotype.

Finally, to further support the conclusions about daily rhythms of feeding behavior, it would be valuable to perform dye assays with either sugar or yeast separately.

We thank the reviewer for this comment. We are very interested in the observation we made that macronutrient appetites likely undergo daily rhythms. However, to accurately test this besides the experiments the reviewer suggested, we would need to do additional ones e.g. to test free-running rhythm in DD or disturb the core clock, which we fell are out of the scope of the current study. As we mentioned, we are very interested in this question, and we aspire to make it part of another study of ours.

One key mechanistic question is whether upd3 is activating JAK-STAT signaling in NPF+ neurons. Have the authors examined the role of the upd3 receptor dome in these neurons? Knockdown of dome in NPF+ cells would provide strong support for this pathway. Similarly, how is ImpL2 acting in this context – does it suppress feeding via an effect on insulin signaling in NPF+ neurons? Addressing these questions would provide a more complete picture of the molecular mechanisms underlying tumor-induced anorexia.

We thank the reviewer for this comment. We have added new experiments indicating that upd3 signaling doesn't directly impact NPF neurons instead ImpL2 signaling is the one silencing NPF neurons. In detail:

1) we assessed whether excessive increase of ImpL2 or upd3 in the hemolymph (fly blood) impacts the activation of NPF neurons using an *ex vivo* brain assay (Fig. 3h, Ext. Data. Sup. Fig. 3g-j). We found that adding hemolymph from flies with excessive increase of ImpL2, inhibited the activity of NPF neurons (Fig. 3h, Ext. Data. Sup. Fig. 3h,3j). However, the activity of NPF neurons was not inhibited when adding hemolymph from flies with excessive increase of upd3 (Fig. 3h, Ext. Data. Sup. Fig. 3g,3j). These data point to ImpL2 and not upd3 being responsible for inhibiting NPF neurons.

2) we tested the levels of *NPF* expression in flies with excessive increase of both ImpL2 and upd3 signaling (without tumor growth, Fig. 3g). We found that *NPF* is significantly decreased (40% decrease), resembling low *NPF* in *Yki^{act}* (Fig. 3g). However, excessive increase of upd3 alone didn't change the expression of *NPF* and excessive increase of ImpL2 alone caused only a moderate 20% decrease of *NPF* expression (Fig 3g). Notably, flies with combined ImpL2 and upd3 excess are anorexic, whereas flies with excess of ImpL2 or upd3 alone do not show significant decrease in food intake (Fig. 3e-f). These data indicate that elevated ImpL2 signaling alone leads to a moderate decrease in *NPF* expression which is not sufficient to reduce feeding, whereas ImpL2 in the presence of excessive upd3 leads to a large decrease in *NPF* expression and anorexia.

3) we knocked down the *Drosophila Jak* homolog (*hop*) in the blood brain barrier (BBB) while the *Yki*-tumor was growing in the gut of male flies (Ext. Data. Sup. Fig. 3e). Reduction of upd3 signaling in the BBB of tumor-induced flies significantly increased food intake (Ext. Data. Sup. Fig. 3e). In addition, in these flies *NPF* expression was subtly increased compared to *Yki*-tumor flies but not significantly restored to control levels (Ext. Data. Sup. Fig. 3f), likely due to remaining ImpL2 signaling that alone reduces *NPF* expression.

4) we knocked down *ImpL2* in the gut of *Yki^{act}* male flies which significantly rescued both anorexia and *NPF* expression, to levels similar to control (Fig. 3b-d).

Taken together, our findings indicate that colorectal tumor-induced ImpL2 is responsible for the decline of neuronal NPF signaling and the role of upd3 is to augment the inhibitory impact of ImpL2 that leads to anorexia. We propose that upd3 assists ImpL2 by disrupting the BBB and increase passage of ImpL2 to the brain.

Specifically, we have revised the manuscript to propose: “*Altogether, our data suggest that excessive increase of ImpL2 drives the reduction of neuronal NPF signaling and that upd3 augments the inhibitory impact of ImpL2, leading to anorexia. We propose that upd3 disrupts the BBB and increases passage of ImpL2 to the brain.*”

Reviewer #3 (Remarks to the Author):

A salient feature of cancer cachexia in patient, mouse and fly models is organ atrophy and weight loss. Although experiments from rodent models early on showed that forced feeding does not fully reverse weight loss and organ atrophy, evidence has amassed that substantiates that anorexia is a rate-limiting component of cachexia progression with atrophy and weight loss. The molecular mechanisms mediating cachexia-anorexia are still largely unclear.

In flies, a seminal paper addressing this question in a fly model <https://doi.org/10.1038/s41556-020-00628-z> showed that a tumor model based on Yki-act overexpression in the eye of adult flies induces anorexia through tumor Dilp8 insulin-like peptide secretion that is sensed by Lgr3 receptors in select (NPF) neurons in the brain. Lgr3 and Dilp8 suppress feeding and reduce orexigenic sNPF and NPF production and increases anorexigenic NUCB1 production. Strikingly, the analogous signaling network and peptides are conserved to experimental mouse cancer cachexia models and pancreatic human tumor cells that express the dilp8 orthologue/INSL3 and affect orthologues of NPY/NPF and NUCB2/NUCB1 feeding regulators. The causal relationship of NPF/sNPF was only explored in mice.

In the submitted manuscript by Petsakou A, et al, utilizes a “colon cancer” Yki-act model driven by the escargot-GAL4 driver of gut stem cells and lineage and investigates effects on feeding and weight loss. Previous findings from the same lab reported no/little effect on feeding, but more sensitive methods now reveal reduced food intake that precedes a reduction in dry body weight. In this model, NPF is reduced, whereas NUCB1 and sNPF were not measurably different in expression at the whole-body level, so the authors conclude that something else may be at play, distinct from eye Yki-act model. The authors chose to focus on NPF neuron regulation and how it mechanistically relates to anorexia & weight loss as this question was not pursued in detail in previous studies.

In a series of impressive experiments combining, sophisticated genetics, gene expression analysis, neuronal activity manipulation, in vitro culture and food preference functional analysis, the manuscript makes a case for a tumor-induced change in food preference that may underlie anorexia and weight loss.

We thank the reviewer for the support!

After establishing that Yki-act gut tumors lead to weight loss, the authors ascribe this. Function to the combined expression of the Insulin growth factor binding protein, IMPL2 and Upd3/IL-6, earlier shown to be expressed in Yki-act tumors. This is based on sufficiency experiments by combined ectopic expression of these factors in muscle that reproduce reduced feeding and weight loss effects. Knockdown of these genes in Yki-act tumors is not shown to address if these factors are required in the tumor setting and whether they derive only from tumor. Whether Dilp8/Lgr signaling is involved or acts entirely separate is also not addressed. If raising NPF levels sufficient/insufficient to rescue the defects is also not experimentally addressed. Instead, the authors show that NPF neuronal activity is a likely mediator. Combined sophisticated imaging and manipulation of NPF neurons' activity show that NPF activity is lower when exposed to hemolymph from tumor-carrying animals and that stimulating NPF signaling in vivo rectifies feeding and food choice defects. Whether IMPL-2 and Upd3 directly affect NPF neurons or whether this is mediated through other secondary effectors or metabolites is not addressed. By manipulating NPF activity, it is shown that the tumor-induced reduced preference for protein rich food can be rectified through increasing NPF neuronal signaling. The downstream mechanisms for this behavioral effect are not addressed.

We thank the reviewer for these comments. Our revised manuscript has now addressed the above points which we outline below in detail as we address the specific questions asked by the reviewer.

Experiments are well executed with relevant controls and statistics according to the standard of the field.

We thank the reviewer for the support!

Altogether, the manuscript invokes reduced NPF neuronal activity in regulating feeding and food choice in response to tumor produced factors. They make a convincing argument that NPF activity, possibly in addition to reduced NPF expression, controls food choice and ingestion, but their relative contribution is not addressed. Open questions remain and need to be addressed

We thank the reviewer for the support! We have addressed the reviewer's questions in detail below

1. Is Dilp8-LGR3 signaling involved in NPF neuron signaling and regulating NPF levels as suggested by the eye Yki-act tumor model? Experiments to include or exclude Dilp/Lgr3 should be shown. I.e. does Lgr3 knockdown in NPY neurons affect neuronal activity/food choice caused by Yki gut tumors? Is Dilp8 increased in Yki-act gut tumor models?

We thank the reviewer for this comment. We have added new experiments that address the reviewer's questions about *dilp8*. Specifically, we tested the levels of *dilp8* in colorectal Yki-tumor flies at several timepoints until the onset of anorexia and organ wasting (from d3 to d6) and we did not observe a significant increase of *dilp8* expression (Ext. Data Sup. Fig. 3a). In addition, we knocked down *dilp8* in the gut of Yki^{act} flies at d6 but we didn't observe a rescue in feeding behavior (Ext. Data Sup. Fig. 3b). Together these data suggest that *dilp8* does not regulate anorexia in the colorectal Yki-tumor model.

2. How does reduced NPF levels relate to NPF neuronal activity and food preference? Are these parallel events, and does NPF levels matter? A rescue of NPF levels in NPF neurons could be experimentally addressed in Yki-act animals. Can the authors disentangle NPF activity from NPF production?

We thank the reviewer for this comment. Given that NPF is a well-studied neuropeptide, this question has been addressed multiple times in previous works. One of the early works that first linked NPF with feeding behavior did so by linking food sensing with increased NPF expression, elevated protein NPF levels, and increased NPF-neuronal transmission (PMID: 11257610). In addition, several studies reported that when NPF expression is reduced then the associated behavior which is dependent on NPF signaling is reduced as well (PMC10743384, PMC5617300, PMC5536846). Also, several studies co-expressed NPF-RNAi while causing TrpA1-induced NPF increase and found that it diminished the associated behavior (PMC9684077, PMC8355161, PMC10743384, PMC5536846, PMC5617300). Using a similar approach, we tried to disentangle NPF activity and production by co-expressing NPF-RNAi in our tumor model while inducing NPF activation. However, it was not possible to generate flies that co-expressed both *TrpA1* and NPF-RNAi under the colorectal tumor-Yki background due to the packed transgenic load on all chromosomes.

We have now referenced the above literature in our revised manuscript to better clarify the link between NPF expression and NPF neuronal activity. We wrote: ".....Moreover, NPF is a well-studied conserved neuropeptide and low levels of NPF reduce associated behaviors including feeding (PMID: 11257610, PMC9684077, PMC5536846, PMC5617300, PMC10743384)..... Specifically, at high temperature (~29°C) *TrpA1* allows the influx of cations inside the cell which increases activation and release of NPF from NPF-expressing neurons and

enteroendocrine gut cells (PMC9684077, PMC8355161, PMC10743384, PMC5536846, PMC5617300)”

3. Upd3-IMPL2 both are needed to target NPF neuronal activity. Whether this is due to direct control or if indirect mechanisms are at play is left unaddressed. Previous work has shown that both Upd1 and S6K status (downstream in Insulin-TORC1 signaling) can regulate NPF neurons and feeding. Addressing if the Upd receptor (Dome) and InR-PI3K-TORC1-S6K signaling affects NPF neuronal activity could be experimentally addressed to see if any of the mediators act directly on the NPF neuron through IMPL2 and Upd3.

We thank the reviewer for this comment. Our new data point to a synergistic function of ImpL2 and upd3 signaling, with ImpL2 signaling efficiently reducing *NPF* and feeding behavior only in the presence of excessive increase of upd3 signaling. In detail:

1) we tested the impact of excessive increase of upd3 signaling in NPF neurons using an *ex vivo* brain assay (Fig. 3h, Ext. Data. Sup. Fig. 3g-j). We found that adding hemolymph from flies with upd3 excess allows NPF neuronal activation (Fig. 3h, Ext. Data. Sup. Fig. 3g,3j). Instead, the activity of NPF neurons is repressed when we added hemolymph from flies with excessive increase of ImpL2 in the hemolymph. (Fig. 3h, Ext. Data. Sup. Fig. 3h,3j). These data point to ImpL2 signaling and not upd3 to be responsible for inhibiting NPF neurons.

2) we tested the levels of *NPF* expression in flies with excessive increase of ImpL2 and upd3 signaling (without tumor growth, Fig. 3g). We found that *NPF* is significantly decreased (40% decrease), resembling low *NPF* in *Yki^{act}*. However, excess of upd3 alone didn't change the expression of *NPF* and excess of ImpL2 alone caused only a moderate 20% decrease of *NPF* expression (Fig 3g). Notably, flies with ImpL2 and upd3 excess are anorexic, whereas flies with excess of ImpL2 or upd3 alone do not show significant decrease in food intake (Fig. 3e-f). These data indicate that elevated ImpL2 signaling alone leads to a moderate decrease in *NPF* expression which is not sufficient to reduce feeding, whereas ImpL2 in the presence upd3 leads to a large decrease in *NPF* expression and anorexia.

3) we knocked down *ImpL2* in the gut of *Yki^{act}* male flies which rescued both anorexia and *NPF* expression, to levels similar to control (Fig. 3b-d).

4) we knocked down the drosophila *Jak* homolog (*hop*) in the BBB while the *Yki*-tumor was growing in the gut of male flies. Reduction of upd3 signaling in the BBB of tumor-induced flies significantly increased food intake (Ext. Data. Sup. Fig. 3e). In these flies *NPF* expression was subtly increased compared to *Yki*-tumor flies but not restored to control levels (Ext. Data. Sup. Fig. 3f), likely due to lingering ImpL2 signaling that alone reduces *NPF* expression.

Together, our findings suggest that in *Yki^{act}* flies ImpL2 signaling is responsible for the reduction of *NPF* expression and NPF neuronal activation. However, for ImpL2 signaling to strongly reduce NPF signaling to levels sufficient to drive anorexia, it requires upd3. We propose that upd3 augments the inhibitory effect of ImpL2 in NPF signaling by disrupting the BBB and allowing aberrant access of ImpL2 to the brain.

4. Why is Upd3/IMPL2 double knockdown not shown for tumor tissue? This is important and would be necessary to argue that the tumor secretes these factors to regulate NPF neurons and in turn regulate feeding. It is possible that ImpL2 and Upd3 expression in response to Yki-Act tumors is broader than the tumor itself, furloughing direct tumor knockdown experiments. If this is the case, this should be showed by using existing reporters (preferred), IF analysis or qPCR. Or is the Upd3/IMPL2 effect thought to act indirectly through another mechanism?

We thank the reviewer for this comment. We have addressed the reviewer comments by knocking down *ImpL2* and *upd3* in the gut of *Yki^{act}* flies (Fig. 3b-d, Ext. Data Sup. Fig. 3b) as well as by showing the elevated expression of *upd3* and *ImpL2* in the gut of *Yki^{act}* flies (Ext. Data Sup. Fig. 3c). Knocking down of *ImpL2* in the gut of *Yki^{act}* flies was sufficient to rescue both

NPF expression and feeding behavior by d5 (Fig. 3b-d), indicating that elevated *ImpL2* signaling from the tumor itself drives *NPF*-linked anorexia.

Even though knocking down *upd3* in the gut of *Yki^{act}* flies rescued anorexia (Ext. data Sup. Fig. 3b), this is due to inhibition of the tumor itself (as previously reported, PMC8410949) rather than a direct effect on feeding. Our data suggest that elevated *upd3* signaling in *Yki^{act}* flies regulates anorexia indirectly by disrupting the BBB (Ext. data Sup. Fig. 3b).

Specific comments:

Fig1 It seems like the esgTs>GFP have higher expression in the males in general (brain and gut). Is it true that there is sex differential esg expression, or is it only a trend in these pictures? Could these explain why the feeding phenotype is more dramatic in the males? .

We thank the reviewer for this comment. Each image was acquired for experiments that tested for differences of *Yki*-tumor flies compared to control. As a result, immunostaining and image acquisition in male flies (*Yki*-tumor vs control) was conducted at different times compared to females (*Yki*-tumor vs control). It is therefore not accurate to test for differences in the images of male versus female flies. We apologize to the reviewer if this was not clear in our initial manuscript. To make this clearer in our revised manuscript, we show all experiments of male flies in Fig.1 and all experiments of females in Ext. Data Sup. Fig. 1.

In addition, using one-way Anova Kruskal-Wallis multiple comparison test we analyzed the anorexic phenotypes of male versus female *Yki*-tumor flies (using data from the FlyPAD experiments of *Yki^{act}* and *esg^{TS}> Yki^{act}* flies). We didn't observe any significant differences in the reduced feeding behavior of male vs female *Yki*-tumor flies (p-values >0.9999). This indicates that anorexia in *Yki*-tumor flies is consistently dramatic in both male and females. We clarify that in our revised manuscript by writing : "*Using the FlyPAD, we observed a significant reduction in the feeding behavior of individual esg^{TS}>yki^{act} flies, with both males and females showing similar decrease in feeding behavior (Fig. 1a, Extended Data Sup. Fig. 1a).*"

No quantifications are shown for tumor size when gut tumors are shown. Can the authors provide tumor volumes for a given gut region to make a firm statement on the effects on tumor growth?

We thank the reviewer for this comment. In the revised figure Ext. Data Sup. Fig. 2b we provide quantifications of GFP levels of the posterior gut as an indicator of tumor growth in *Yki^{act}* male flies.

Fig2 a-b. Why is whole body expression assessed for ImpL2 and upd3 rather than from tumor/gut? From previous work, the expectation is that these secreted proteins originate for tumor cells. Do the authors expect or observe otherwise? Reporters for Upd3 and IMPL2 exist. Where are they expressed in control and tumor conditions? Is their expression controlled by starvation?

We thank the reviewer for this comment. In our revised manuscript we added qPCR experiments from the gut showing that by d4 of colorectal *Yki*-tumor growth *ImpL2* and *upd3* expression is significantly elevated in the gut (Ext. Data Sup. Fig. 2c).

Previous work reported that upon colorectal *Yki*-tumor growth *ImpL2* is highly expressed from the gut and other tissues (like the muscle) and together all contribute to the elevated secretion of *ImpL2* in the hemolymph during tumor growth (PMC8410949). Due to this study, we considered more accurate to test the levels of tumor secreted factors like *ImpL2* in the whole body of flies.

In addition, *ImpL2* expression is elevated during starvation (PMC2323038). However, in colorectal *Yki*-tumor flies, excessive increase of *ImpL2* in the gut and whole body starts at d4 (Ext. Data Sup. Fig. 2c, Fig 3a), one day prior to the onset of anorexia at d5. Therefore, our data

support that increase of *ImpL2* expression at d4 is due to colorectal tumor growth (PMC4437243) and not to anorexia and subsequent starvation which start at d5.

2d-e. While it is interesting that the combined effect of ImpL2 and Upd3 expression from muscle, lead to reduced feeding, it is unclear why this approach is chosen. Why not express using Mex-GAL4 (gut) for instance, which approximates the tumor model in location and possibly cell type(s). Are Upd3 and ImpL2 expressed in muscle upon 1. gut tumor presence or 2. starvation?

We thank the reviewer for this comment. We have changed the length to a full article and have now clarified in the manuscript why we chose to overexpress *ImpL2* and *upd3* in the muscle.

We used this approach as it was previously used by the Bilder lab (PMC4390765) to specifically assess the impact of a secreted factor (*ImpL2*) in organ wasting while avoiding the background effect of the tumor growth. In this study, large tissues like the muscle were chosen so that the secreted factor would be released in excess in the fly hemolymph (blood), reminiscent of the high amounts during tumor growth (PMC4390765). The advantage of this approach is that you can particularly assess what is the impact of a specific signal without the indirect effect of additional tumor-induced factors and tumor-induced complications.

Another reason why we chose the muscle as a large tissue is because previous work by the Song lab reported that during colorectal *Yki*-tumor growth *ImpL2* is released from the gut but is also supplementary released from the muscle which amplifies the excessive amount *ImpL2* secreted in the hemolymph (PMC8410949).

We have now revised our manuscript like this: *“Previous work tested the role of ImpL2 in organ wasting independent of tumor growth by ectopically over-expressing ImpL2 in large tissues such as the muscle, and observed that excessive circulation of ImpL2 in tumor-free flies is sufficient to promote organ wasting (PMC4390765). To test for a role of ImpL2 in anorexia independent of colorectal tumor growth, we took a similar approach and conditionally overexpressed ImpL2 from a large tissue. We chose the muscle because it was used previously (PMC4390765) and because in colorectal Yki-tumor flies elevated amounts of circulating ImpL2 originate primarily from the gut but have also been reported to come from the muscle (PMC8410949).”*

To make a firm case that tumor-produced Upd3 and IMPL2 mediates anorexia, the authors should provide experiments where these factors are knocked down in tumors. The reduced growth of tumors upon Upd3 knockdown can be counteracted by Hop-Act in tumors for instance if reduced tumor growth is a caveat. Epistasis experiments in NPF neurons to show if Upd3 and IMPL2 acts directly on NPF neurons could be addressed by manipulating Dome and InR-S6K signaling. Why are these experiments not pursued.

We thank the reviewer for this comment. In our revised experiment we have added experiments where *ImpL2* and *upd3* is knocked down in the gut in *Yki^{act}* flies at d5 and d6 (Fig. 3b-d, Ext. Data Sup. Fig. 3b). We have also added epistatic experiments where *upd3* signaling is reduced in the blood brain barrier (BBB) by knocking down the *Drosophila Jak* homolog (*hop*) in colorectal *Yki*-tumor flies at d5 (Ext. Data. Sup. Fig. 3e-f). Together these experiments point to a synergistic role of *ImpL2* and *upd3* signaling that drives NPF-linked anorexia in *Yki*-tumor flies.

Based on these experiments the mechanism we propose in the revised manuscript is: *“Altogether, our data suggest that excessive increase of ImpL2 drives the reduction of neuronal NPF signaling and that upd3 augments the inhibitory impact of ImpL2, leading to anorexia. We propose that upd3 disrupts the BBB and increases passage of ImpL2 to the brain.”*

Fig3

In the previous eye-Yki-act model studies (reference 37), they used females to study anorexia. Did the authors address if this is the reason for differences in sNPF and nucb1 between the studies?

We thank the reviewer for this comment. In the above-mentioned study (previous ref. 37, PMID: 33558728) the authors use both male and female flies depending on the experiment they are conducting. For example, food intake assay was conducted in male flies, while TAG assay was conducted in females and in some other assays they do not clarify (PMID: 33558728).

In our study, we tested the levels of *sNPF* and *nucb1* in male (Fig. 2a) and female (Ext. Data Sup. Fig. 2a) colorectal Yki^{act} flies and did not observe any significant difference in either. We also tested in colorectal Yki-tumor male flies the levels and role of *dilp8*. We didn't observe any increase in *dilp8* and knocking down *dilp8* in the gut didn't rescue the anorexia in Yki^{act} flies by d6 (Ext. Data Sup. Fig. 3a-b). Altogether our findings suggest that the underlying mechanism driving anorexia in the colorectal Yki-tumor model is different from the mechanism proposed in the eye-tumor model.

g – Why are there so many dots on the graph that are on 0? How long were these bouts? Should there be a filter to what is considered a bout?

We thank the reviewer for this comment. The FlyPAD is an automated feeding assay, that records every time the fly is in contact with the food. The duration that the fly is in contact with the food correlates strongly with food intake (PMC4143931). Flies that are not in contact with the food and therefore do not eat are recorded as 0.

Activity bouts are automatically calculated and filtered as described by Itskov et al (PMC4143931). Specifically, the FlyPAD records when the fly is in contact with the food in consecutive 500ms windows. A filter is already applied because a signal is recorded only when it is above a fixed background threshold. Activity bouts are defined as epochs that the signal surpasses that fixed threshold (PMC4143931). The FlyPAD metrics we use is the total duration of activity bouts (in seconds) which is the cumulative time the fly is in contact with the food and is reported to be most strongly correlated with food intake (PMC4143931).

g- Why are they never seen or seen very little on the control column in all flypad graphs?

We thank the reviewer for this comment. As we describe in the Method section (Feeding Assays) we test overall food intake for 1-hour between ZT0-ZT2 (2 hours from when lights are on). We assay food intake during ZT0-ZT2 because it is when control flies eat significantly more compared to other times throughout the day (PMC2703740). So, during the 1-hr window between ZT0-ZT2, most control flies will eat and come in contact with the food.

Moreover, by the end of all FlyPAD assays we ensured that assays were conducted properly by checking the state of the food and flies in each behavioral arena. If the food was dried out completely, or flies were injured or dead or had escaped during their transfer to the arena we discarded these experiments.

We have revised that our manuscript to better clarify these by adding : *“All flies were trained in 12:12 (Light: Dark) cycles at 29°C (unless flies were assayed on d0 where they remained at 18°C), before performing behavioral experiments.....We monitored the feeding activity (total duration of activity bout, in seconds, PMC4143931) of individual flies in the well with the lab food for 1hr at RT between ZT0-ZT2 (12:12 LD), the time of day that control flies eat significantly more (PMC2703740)..... After each FlyPAD assay we checked the state of food and flies in each behavioral arena. If by the end of an assay the food had completely dried out or a fly was injured or dead or had escaped the arena, then these experiments were discarded.”*

g- Why is there a difference with the third group? If there is no tumor, and there is no contribution from the gut NPF, why do these flies eat less?

We thank the reviewer for this comment. Previous studies support that NPF signaling from the brain impacts food intake inversely to NPF signaling from the gut (PMC5536846, PMC9684077, PMC8355161). Specifically, TrpA1-induced NPF activation from the brain promotes feeding (PMC5536846) but from the gut it inhibits feeding (PMC9684077). In addition, knocking down *NPF* solely in the gut leads to an increase in food intake (PMC9684077, PMC8355161).

We apologize to the reviewer for not making this distinction clearer in the previous version of our manuscript. We now clarify this in our revised manuscript, by

- 1) adding in the introduction: “*In addition, NPF is found in enteroendocrine cells in the fly gut (PMID: 10499420), where is reported to have distinct physiological functions from the brain, e.g. NPF signaling from the gut promotes sugar satiety and inhibits food intake (PMC9684077, PMC8355161).*”
- 2) adding in the result for section for Fig. 2g (previous Fig. 3g): “*In addition, flies with TrpA1-induced activation of NPF solely from the gut showed reduced food intake (Fig. 2g), in agreement with previous reports that NPF signaling from the gut has distinct roles from the brain (PMC9684077, PMC8355161).*”

h- Why do the authors study NPF neuronal activation at D6 and not at D4, as the expression drop was found at D4 in fig. 3b

We thank the reviewer for this comment. In our preliminary experiments we tested hemolymph at d4, d5 and d6 of *Yki^{act}* flies and observed a reduction of NPF activation in all. However, we chose d6 because we thought that it would include higher levels of tumor-secreted factors.

Fig4

a – It would be favorable if the authors could provide tumor volumetrics.

We thank the reviewer for this comment. As indicator of tumor volumetrics, we assayed the levels of GFP which is expressed by the *esg-Gal4* driver in the gut of control (*elavGal80;esg^{TS}>GFP*) and *Yki^{act}* (*elavGal80;esg^{TS}>GFP+ Yki^{act}*) flies, including flies with TrpA1-induced NPF activation (Fig. 2e, Ext. Data Sup. Fig. 2b, previous Fig. 4a). Flies carrying colorectal *Yki*-tumors had significantly elevated levels of GFP independent of NPF activation (Fig. 2e, Ext. Data Sup. Fig. 2b).

Optional/suggestion:

- Survival curves for rescues of anorexia. It would be interesting if there is a link between NPF neuronal activity rescue and whether it leads to increased animal survival through reversal of anorexia. This may, however, be beyond the scope of this work.

We thank the reviewer for this comment. We have added these experiments in Fig. 5c-d of the revised manuscript as the reviewer requested. Indeed, we find a significant improvement in the survival and risk of death of colorectal *Yki*-tumor flies when anorexia is rescued due to early increase of NPF signaling. Specifically, TrpA1-induced increase of NPF in colorectal *Yki*-tumor flies led to significant increase in median and overall lifespan compared to *Yki^{act}* flies, as well as a significant decrease in the probability of death between d21-d24, which we tested because is the approximate half-life of a control fly (~23 days, Fig. 5c-d).

We thank all reviewers for their support!

We have now updated the revised manuscript as requested by reviewer #1 and the editor. The new revisions do not change the conclusions of our study. Please find below our detailed answers.

Reviewer #1 (Remarks to the Author):

The updated manuscript has undergone substantial revision, strengthening the previous findings, providing additional mechanistic insight into the synergy between ImpL2 and upd3, and adding new survival data. The authors have expanded the female dataset to justify their focus on males. These changes significantly improve the manuscript and address the reviewer's earlier concerns. However, several major issues remain to be resolved.

Main Points

1. This point should have been raised in the first review round but was inadvertently omitted from my summary. There is a concern that the authors overly "humanize" the fruit fly model. While the reviewer supports efforts to emphasize the relevance of Drosophila-based research, in some cases the interpretation extends too far. Specifically, referring to experimentally induced midgut neoplasia as "colorectal tumors" requires reconsideration. Although the midgut is part of the fly gastrointestinal tract, it is not functionally or morphologically equivalent to the mammalian colon or rectum. The reviewer understands the authors' intent to engage the medical community; however, equating the midgut with the colon is misleading. The authors are encouraged to provide a clear and defensible justification for using the term "colorectal" in the context of their study.

We thank the reviewer for the comment, and we have now edited the language in our manuscript. Specifically: i) we removed from our title the word "*colorectal*" and changed our title to "*Tumor-induced orexigenic imbalance lowers protein appetite and drives early organ wasting symptoms*"; ii) we replaced in our manuscript the word "*colorectal*" with "*gut*". This change can be found in the manuscript with red letters

Similarly, when discussing the role of blood-brain barrier weakening in enabling neuronal expression of insulin-like hormones during cancer cachexia, the authors should include a disclaimer acknowledging that this mechanism is specific to insects and other invertebrates and cannot be directly extrapolated to vertebrates, in which insulin and insulin-like peptides are produced by non-neuronal tissues. Such transparency will be valuable to readers navigating the complex and sometimes inconsistent literature on cancer cachexia.

We thank the reviewer for the comment. We have now clarified and revised our manuscript by writing (red letters in the manuscript):

i) "*In addition, insulin signaling (from insulin-producing cells in the brain) and upd-related signaling from the brain and fat body are part of the satiety-promoting network in Drosophila that stops feeding and represses orexigenic signaling including NPF (PMCID: PMC6451361, PMC5235317, PMC3475207, PMC1201572).*

ii) "*One of the reported functions of upd3 and IL-6 in fly and mouse tumor models respectively is to disrupt the blood brain barrier (BBB) (PMCID: PMC8511098).*

2. The conclusions drawn from Fig. 5c–d require careful reconsideration. The mortality differences presented at 50% and 0% survival (Fig. 5c) are not as clear as implied. Additional

checkpoints at 25% and 75% survival would likely show that the differences between the $elavGal80;eagTS>yki; \pm TrpA1$ groups are not statistically significant.

We thank the reviewer for the comment. The survival curves shown on Figure 5c are generated using GraphPad prism including the statistics. Specifically, the statistical differences in Figure 5c are not solely for specific survival points but for survival curves as a whole per genotype. As we described in our methods (section statistical tests) we compared the survival curves using the Log-rank (Mantel-Cox) test that is recommended by GraphPad prism and determines if survival curves are significantly different between them. Using the Log-rank (Mantel-Cox) test the p-value between $elavGal80;esg^{TS}>yki^{jact}$ and $elavGal80;esg^{TS}>yki^{jact} +NPF$ -activation ($TrpA1$) is 0.0421, indicating that they are significantly different.

We apologize to the reviewer if the way we previously showed the statistical significance was confusing. We have now revised Figure 5c to clearly show that we are comparing the survival curves, and we are also now specifying it in the legend. (We wrote in the legend for Figure 5c : “*p-values comparing survival curves per genotype were obtained with the Log-rank (Mantel-Cox) test*”). In addition, we have revised the table to add more points and now show the day that each survival curve passes the 75%, 50% , 25% and 0% survival point. We have moved this table to Ext. Data Sup. Fig. 5c. We have also submitted the raw data for this figure in the Source Data excel spreadsheet.

In Fig. 5d, the comparison between the two rightmost groups (i.e., $elavGal80;eagTS>yki; NPFlexA>+$ and $elavGal80;eagTS>yki; NPFlexA>TrpA1$) cannot reasonably yield a p-value < 0.001 as indicated by the triple asterisks. Although the reviewer does not have access to the raw data, simulating datapoints from the plotted values produces a Student's t-test p-value > 0.05. Given these limitations, presenting NPF alterations as the primary explanation for improved survival appears premature.

We thank the reviewer for the comment. The probability of death in our previous Fig. 5d consisted of data for d21 and d24 grouped together. We apologize to the reviewer for not explaining more clearly that these data were grouped.

In detail, there were 18 data per genotype, shown as 9 datapoints as each of the 9 datapoints was the average of d21 and d24 from the same biological sample (9 in total) per genotype. This average datapoint per biological sample and genotype was generated automatically by GraphPad Prism. The reason why we decided to group our data for d21 and d24 is because we thought that this average point between d21-24 per biological sample would be more closely representative of the approximate percentage of death by d23 (which is the half-life of a control fly).

The recommended statistical test for grouped data is two-way Anova using Tukey's multiple comparisons test, which is the test we previously conducted (as we described in our previous legend for Fig. 5d.) Since two-way Anova using Tukey's multiple comparisons test takes into account that each genotype consists of 18 data (grouped into 9), this test found that $elavGal80;esg^{TS}>yki^{jact}$ vs $elavGal80;esg^{TS}>yki^{jact} +TrpA1$, were significantly different with p-value 0.0005.

We have now revised Figure 5d and show the data for day 21 and day 24, not as grouped data, but instead as individual data. We compared these data with one-way Anova using Tukey's multiple comparisons test. Our conclusions (and data) remain the same as the risk of death between $elavGal80;esg^{TS}>yki^{jact}$ vs $elavGal80;esg^{TS}>yki^{jact} +NPF$ -activation is significantly different (p-value = 0.0078) during the half-life of a control fly. We have revised the legend by adding that: “*Boxplots show the probability of death for each of the 9 biological samples on day 21 and for each of the 9 biological samples on day 24 per genotype*”. We have also submitted the raw data in the Source Data excel spreadsheet.

3. Interpretation of male vs. female datasets. Now that comparable datasets are available, the female data do not appear substantially different from the male data. Both NPF and ccha2 expression in females show the same directional trends as in males, even if not highlighted as significant. As the authors note in their rebuttal, adjusting statistical methods could yield significance for factors that visually differ between control and yki+ conditions. In fact, aside from the increase in upd2, few features distinguish the female dataset. In this context, the authors' insistence on treating males separately seems insufficiently supported. A more straightforward solution may be to combine the male and female datasets into a unified analysis, which would simplify interpretation and eliminate the added complexity of sex-specific handling.

We thank the reviewer for the comment. As recommended by the editor, nature communications prefer male and female data to be shown separately, which is why we have kept our figures as is.

In addition, Upd2 the *Drosophila* homologue of Leptin is a key secreted factor from the fat tissue that regulates feeding and energy storage by influencing insulin signaling (PMC3475207). So, the fact that *upd2* prior to organ wasting exhibits significant expression changes only in females, is strong indication on its own that at least feeding (and perhaps even organ wasting) is regulated differently between male and female flies. Therefore, we believe that a distinction on male and female datasets is required even on the basis of *upd2* alone.

However, it is not only *upd2* different between male and female gut Yki-tumor flies on d5. To better visualize the trends between the different datasets in male and female we have added Table S1 (See Supplementary information, Table S1 is at the end of the revised manuscript) which depicts the statistical differences after using Sidak's and after using Uncorrected Fisher's LSD multiple comparisons test. As we explain in the legend of Table S1, we show only Sidak's test in our figures because it is less likely to include false positives. We also revised our manuscript to reference Table S1, so that the trend of the feeding-related signals that are changing expression is clearer to the reader.

Minor Points

4. Add male/female designations to all supplementary figure panels (Figs 1, 2, supple 1, and supple 2), as done in Supplementary Figure 2a, to prevent confusion for readers encountering similar datasets across different figures.

We thank the reviewer for the comment. We have now added male and female designations in all figures as requested. We have also highlighted in every figure legend with blue lettering the male and female designations.

5. The Methods section does not specify how the flies were aged for the experiment shown in Fig. 5d. Please include this information.

We thank the reviewer for the comment. We have now revised our lifespan method section as such: "Newly enclosed flies were collected weekly (Monday-Tuesday) and kept at 18oC (the same temperature that were grown). Lifespan experiments started Fridays and were conducted at 29°C in 12:12 (Light-Dark) cycle conditions. Male flies were transferred to new standard lab food vials every 2-3 days (every Monday, Wednesday and Friday) during light hours and death counts were recorded."

6. Line 382: the word "be" is missing and should be added.

We thank the reviewer for the comment, we have now added the missing word. (see in red)

Reviewer #2 (Remarks to the Author):

The authors have adequately addressed my previous concerns. The added epistatic and mechanistic experiments, together with clearer methodological justification, substantially strengthen the manuscript.

We thank the reviewer for the support!

Reviewer #3 (Remarks to the Author):

The revised manuscript by Petsakou A et al. has been considerably reworked and improved. The authors addressed all my concerns to my satisfaction.

We thank the reviewer for the support!

REVIEWERS' COMMENTS

Reviewer #1 (Remarks to the Author):

Thank you to the authors for addressing my points and providing thoughtful responses. While some data — particularly those related to survivorship — remain open to interpretation, the reviewer concurs with the authors' position and do not wish to press these points further.

We thank Reviewer #1 for the support and agreement.